# Detective SAM: Adaptive AI-Image Forgery Localization

**Gert Lek[1] Nicolas Van Schaik[2] Chaoyi Zhu[2] Pin-Yu Chen[3] Robert Birke[4] Lydia Y. Chen[1,2]**

[1]University of Neuchâtel      [2]Delft University of Technology
[3]IBM Research      [4]University of Turin
gert.lek@unine.ch

## Abstract

Image forgery localization in the generative AI era poses new challenges, as modern editing pipelines produce photorealistic, semantically coherent manipulations that evade conventional detectors while model capabilities evolve rapidly. In response, we develop `Detective SAM`, a framework built on SAM2, a foundation model for image segmentation that integrates perturbation-driven forensic clues with lightweight feature adapters and a mask adapter to convert forensic clues into forgery masks via automatic prompting. Moreover, to keep up with the rapidly evolving capabilities of diffusion models, we introduce AutoEditForge: an automated diffusion edit generation pipeline spanning four edit types. This supplies high-quality data to maintain localization accuracy under newly released editors and enables up-to-date periodic fine-tuning for `Detective SAM`. Across four benchmark datasets and seven baselines, Detective SAM delivers stable out-of-distribution performance, averaging 34.68 IoU / 42.03 F1, a **38.94%** relative IoU gain over the best baseline. Further, we show that state-of-the-art edits cause localization systems to collapse. With 500 AutoEditForge samples, `Detective SAM` quickly adapts and restores performance, enabling practical, low-friction updates as editing models improve. The pretrained weights, AutoEditForge, and evaluation script are available at the GitHub repository.

## 1 Introduction

Deep learning has democratized photorealistic image generation. Synthetic images from modern models are often indistinguishable (Ramesh et al., 2021) to the human eye. Targeted edits can change identities, alter evidence, and mislead viewers even when the rest of an image is authentic (Kadha et al., 2025). As our virtual environment floods with such content, there is an urgent need to identify where an image has been altered. Image forgery localization (IFL) can be challenging in the context of modern local editing, where small, realistic insertions and removals frequently evade human perception. Figure 1 displays such an edit from NanoBanana (Gemini 2.5 Flash Comanici et al. (2025)) and the predictions.

Legacy IFL targeted splicing and copy-move operations (Kwon et al., 2021). Using forensic clues, which are signals leveraged for edit detection and localization, they detect cross-image merges and within-image duplicates. Powerful modern generators, including diffusion models like DALL-E, render legacy clues and methods outdated (Ramesh et al., 2022; Zhang et al., 2024). By design, legacy IFL relies on camera or compression artifacts, which modern generator edits lack because their artifacts are from the generative process (Kwon et al., 2021; Guillaro et al., 2023). New diffusion datasets reveal significant localization drops (Nguyen et al., 2024; Zhang et al., 2024); rapid progress in generative models creates a moving target that requires up-to-date data and training.

This paradigm shift, brought on by diffusion models, initiated a surge in research on stronger forensic clues. Part of this surge shows empirical success with training-free (Ricker et al., 2024; Tsai et al., 2024a; He et al., 2024) and zero-shot (Cozzolino et al., 2024) methods that rely on explicit perturbation artifacts in the embedding space of foundation models. Image foundation models learn embeddings through large-scale self-supervision (Dosovitskiy et al., 2021; Oquab et al., 2024). Such

embeddings reveal distribution shifts in diffusion outputs under perturbations such as Gaussian noise or blur, providing a strong forensic clue for diffusion edits. The recently released Segment Anything

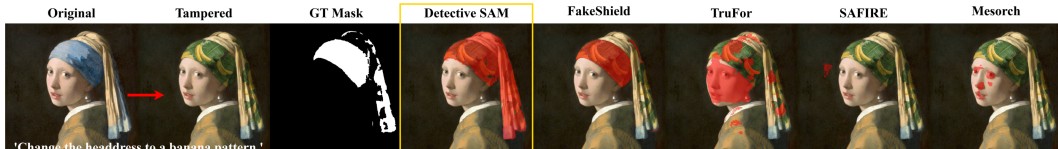

Figure 1: For a NanoBanana sample: original image, tampered image, Ground-Truth (GT) mask, Detective SAM (ours) and baseline mask predictions, limited to models that produced output, for all models see Appendix B.1.

Model (SAM Kirillov et al. (2023), SAM2 Ravi et al. (2024)) serves as a domain-specific foundation model for image segmentation, using a strong, large-scale pretrained encoder. Downstream task performance of SAM has been outstanding (Chen et al., 2024), with applications to shadow & camouflage detection (Jie & Zhang, 2023; Meeran et al., 2024) and IFL. In IFL, SAM is redirected from object to forged-region segmentation. Applications of SAM to IFL are still emerging: current methods (Kwon et al., 2024; Zhang et al., 2025) tend to emphasize legacy forgery methods and neglect more diffusion-specific clues.

Three persistent problems hinder current IFL systems: 1) Current approaches typically avoid using forensic clues that characterize modern edits, failing to leverage the prior information embedded in foundation models. 2) Model architectures should support efficient integration of fresh edited data as it appears, adapting efficiently and avoiding catastrophic forgetting; and, 3) systems must stay effective on recent strong editors, but our experiments show consistent drops on newly released models, indicating a need for continually refreshed training and evaluation data.

In response, we propose `Detective SAM`, a practical framework for modern IFL that addresses these challenges. Building on the insight that the large-scale pretrained SAM2 encoder can detect shifts in the embedding distribution, Detective SAM converts this perturbation-driven forensic clue into an automatic heatmap prompt for SAM2, addressing 1). Through lightweight feature adapters (Chen et al., 2024), SAM2's decoder is retargeted from object segmentation to forgery localization. The backbone of SAM2 remains frozen, and only our modules are trained, mitigating forgetting and enabling efficient, lightweight fine-tuning with replay as new editors appear, attending to 2). Figure 2 summarizes the architecture and SAM2 interactions.

Finally, we directly operationalize challenge 3) via AutoEditForge, ensuring that training and evaluation data remain current. AutoEditForge is an automated pipeline that produces human-like local generative edits of real images with pixel-accurate masks across Replace, Remove, Add, and Change Partially edit methods. It is symbiotic with `Detective SAM`: AutoEditForge supplies fresh edited–real image pairs that enable both evaluation and rapid adaptation.

Our contributions to IFL on generative edits are as follows:

1. **`Detective SAM` architecture** We extend SAM2 for the image forgery localization task with (i) perturbation-driven feature embeddings as a forensic signal, (ii) lightweight adapters that specialize the SAM2 decoder for forged-region segmentation, and (iii) a learnable prompt module that maps the embeddings to a heatmap prompt guiding SAM2 to localize forgeries automatically.

2. **`Detective SAM` for fine-tuning & evaluation** AutoEditForge, an automated pipeline for instruction-based local edits (Replace/Remove/Add/Change Partially), keeps data current as editors evolve and enables up-to-date periodic fine-tuning and evaluation. Coupled with `Detective SAM`'s adapters, designed for efficient fine-tuning, this yields quick recovery of metrics like IoU/F1 on new editors while preserving prior performance.

3. **Comprehensive evaluation** Detective SAM is benchmarked on eight datasets across seven baselines, delivering strong and stable Out-Of-Distribution (OOD) results, yielding a 38.94% gain in average OOD IoU relative to the best baseline. We demonstrate that localizers collapse on recent diffusion edits, necessitating constant fine-tuning.

## 2    RELATED WORK

**Image forgery localization.** IFL concerns itself with the task of not only detecting if parts of an image are manipulated, but also pinpointing them pixel-wise. An effective signal or "forensic clue" is required to locate image forgery. These clues/artifacts can include reconstruction error (Vesnin et al., 2024), JPEG compression artifacts (Kwon et al., 2021), explicit noise artifacts (Zhu et al., 2024a), or implicit noise artifacts (Zhang et al., 2025). Implicit noise artifacts are trained networks that extract specific artifacts from images, such as Noiseprint (Cozzolino & Verdoliva, 2018; Guillaro et al., 2023). In contrast, explicit noise artifacts process features from perturbations without retraining.

Recent work has shown explicit noise artifacts in the embedding space of foundation models. RIGID (He et al., 2024) and BLUR (Tsai et al., 2024a) show that it is possible to detect synthetic diffusion model images using the DINOv2 (Oquab et al., 2024) image foundation model in a training-free manner by detecting subtle embedding distribution shifts. The empirical results show that explicit artifacts appear promising for diffusion model forgery localization/detection. Traditional localization models typically use implicit noise artifacts for copy-move and splicing forgeries, (Kwon et al., 2021; Liu et al., 2022; Guillaro et al., 2023). These methods work well under traditional forgeries, as implicit noise artifacts can effectively capture the compression/camera artifacts of the forged source image. A new branch of IFL using Multi-Modal-Large-Language-Models (MLLMs) arose with models such as SIDA (Huang et al., 2025) and FakeShield (Xu et al., 2025). These methods leverage the text-to-image nature of diffusion model edits to localize forgery and provide explanations.

**SAM in IFL.** Adaptations of SAM for IFL have attracted considerable interest (Kwon et al., 2024; Lai et al., 2023; Zhang et al., 2025). These methods seek to distinguish manipulated regions from genuine content by training SAM to segment forged areas in contrast to the conventional object segmentation task. For example, SAM is adapted for deepfake localization (Lai et al., 2023) with a reconstruction-error signal or used in multi-source forgery partitioning (Kwon et al., 2024) with large-scale contrastive pretraining and a fixed $16 \times 16$ point grid. However, diffusion-based tampering often manifests itself in subtle artifacts and highly irregular regions. Therefore, we require learnable prompts that dynamically adjust to the unpredictable patterns of diffusion-based forgeries. IMDPrompter (Zhang et al., 2025) achieves this with a learnable heatmap and box prompts employing various filters/views as the signal. This technique neither uses an explicit perturbation-driven signal nor builds upon the strong SAM adaptation results from Chen et al. (2024). Therefore, they retrain SAM2's mask decoder. Chen et al. (2024) demonstrate robust downstream performance in camouflage, shadow and medical image segmentation via lightweight feature adapter fine-tuning. Other approaches use SAM's segmentation capabilities without learnable prompts (Su et al., 2024).

**Diffusion dataset generation.** IFL dataset generation has evolved from manual mask and edit prompting (Jia et al., 2023), to using crowd-workers (Zhang et al., 2024), and, at present, fully-automatic dataset creation (Huang et al., 2025; Xu et al., 2025). These fully-automatic pipelines are limited in diverse editing operations like Replace, Remove, Add, and Change Partially, and typically do not employ the most recent diffusion models. Appendix C compares representative pipelines.

## 3    DETECTIVE SAM

We consider the task of image forgery localization, where given an RGB image $\mathcal{I} \in \mathbb{R}^{3 \times H \times W}$ with three channels, height $H$ and width $W$, we aim to predict a binary mask $\mathcal{B} \in \{0, 1\}^{H \times W}$, with $\mathcal{B}_{ij} = 1$ if pixel $(i, j)$ has been edited/tampered, and $0$ otherwise. This work strictly focuses on edits generated by diffusion-based image-editing pipelines. A diffusion model processes an instruction to generate local edits of a source image, as in Figure 1. Keeping the area around the edit unchanged involves overwriting the latents within the mask or injecting noise only inside the masked area (Wu et al., 2025; Lugmayr et al., 2022).

### 3.1    OVERVIEW

Detective SAM augments SAM2 (Ravi et al., 2024) with a perturbation-driven feature stream and lightweight adapters while keeping the backbone frozen. Feature adapters fine-tune SAM2's decoder, and the mask adapter prompts the decoder. This aligns both the decoder and its input with

the forgery localization task. The architecture involves: ① creating perturbed embedded features; ② correcting the original feature with the perturbed ones using Feature Adapters; and ③ all features are then used to create a forensic heatmap prompt with the Mask Adapter, with the respective steps visualized in Figure 2. We next describe the process in more detail, for an overview of notations, see Appendix A.

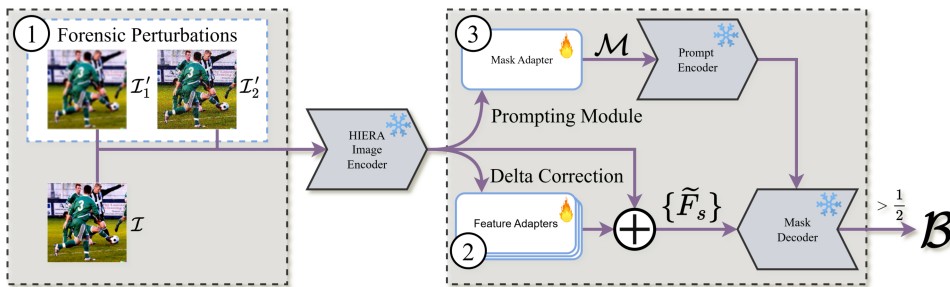

Figure 2: Flow chart of the steps in Detective SAM with our 🔥 learnable modules and pipelines in white and SAM2's ❄ frozen modules in gray. With input image $\mathcal{I}$, perturbed images $\mathcal{I}'_i$, heatmap prompt $\mathcal{M}$, adapted features $\{\tilde{F}_s\}$ and binary forgery mask $\mathcal{B}$. See Appendix B.2 for a flowchart of the original SAM2 components.

## 3.2 MODEL ARCHITECTURE

We build on SAM2 (Ravi et al., 2024), a promptable image/video segmenter featuring a HIERA image encoder (Ryali et al., 2023) producing embeddings at three spatial scales, a prompt encoder for points, boxes, or heatmaps, and a mask decoder that inputs prompts and multi-scale features. The SAM2 backbone (HIERA encoder, prompt encoder, mask decoder) remains frozen. SAM2 was chosen as the backbone because it has a powerful encoder and a promptable decoder that can be adapted due to the joint encoder-decoder training.

Our lightweight adapters are trained jointly, yielding: (i) Three feature adapters (for all three HIERA scales), which input the perturbed image embeddings as a forensic clue and perform a $\Delta F_s$ correction to output the adapted features $\{\tilde{F}_s\}$; and, (ii) A mask adapter, which consists of an automatic prompting network, producing a heatmap $\mathcal{M}$ for the decoder. The feature adapters are single convolutional layers, and the mask adapter contains a transformer that operates in a downscaled embedding space, keeping the model's parameter count modest: with layer width 64, the feature adapters use 81k parameters and the mask adapter 887k parameters. This implies that the model can be trained in two hours on an NVIDIA H100 GPU. Training and inference efficiency are critical for deployment. IFL systems deployed on a platform or on the consumer side need to localize accurately in environments with limited resources. The rapid advancement of diffusion models requires frequent fine-tuning, necessitating training efficiency.

**Inputs and encoding.** As a first step, we construct the forensic feature embeddings for our adapter modules. Given an input image $\mathcal{I}$, we create $N$ perturbed images $\mathcal{I}'_i = \text{Perturb}_i(\mathcal{I}; \theta)$ using simple image-space operators $\text{Perturb}_i(), i = 1, ..., N$ (e.g., Gaussian blur, Gaussian noise, and JPEG compression) with perturbation parameters $\theta$. Diffusion models show embedding shifts under such perturbations (He et al., 2024; Tsai et al., 2024b). Detective SAM leverages these as forensic clues in the form of a localization prior to generative artifacts.

Both $\mathcal{I}$ and $\mathcal{I}'_i$ are encoded by the frozen SAM2 HIERA encoder (Ryali et al., 2023) to produce embeddings $\{F_s^{\mathcal{I}}, F_s^{\mathcal{I}'_i}\}$ at hierarchical scales ($S = \{32, 64, 128\}$) at $(H, W) = (512, 512)$ resolution. To match the image format expected by SAM2's decoder, we pad using SAM2's frozen no-memory (image) embedding, at $s = 32$; and use SAM2's frozen convolutional processing layer $F_s = \text{ConvSAM}(X_s)$ for $s \in \{64, 128\}$. For brevity, without loss of generality, we restrict ourselves to a single perturbation $\mathcal{I}'_1$, yielding six feature embeddings $\{F_s^{\mathcal{I}}, F_s^{\mathcal{I}'_1}\}$ across scales.

**Feature adapters (delta correction).** Next we correct the basic feature embeddings $\{F_s^{\mathcal{I}}\}$ using the forensic perturbed embeddings $\{F_s^{\mathcal{I}'_1}\}$ so that the decoder focuses on forgery localization rather

than generic object segmentation. We achieve this with lightweight feature adapters $\{\mathcal{A}_s\}$ that input the concatenated basic and perturbed $\{F_s^{\mathcal{I}}, F_s^{\mathcal{I}'_1}\}$ to produce a residual delta correction. The $\Delta F_s$ corrections are used to adapt the unperturbed features via a residual connection to produce features;

$$\widetilde{F}_s = F_s^{\mathcal{I}} + \Delta F_s, \qquad \Delta F_s = \mathcal{A}_s\big(\{F_s^{\mathcal{I}}, F_s^{\mathcal{I}'_1}\}\big),$$

which are injected into SAM2's decoder, following the architecture of Chen et al. (2024). The feature adapters are single-layer $1 \times 1$ convolutional networks and specialize the frozen SAM2 decoder to the downstream IFL task with minimal overhead. We provide examples of the learned feature $\Delta$ corrections through a saliency map in the Appendix B.4.

**Mask adapter (automatic prompting).** With the decoder specialized to the IFL task, we replace SAM2's manual user prompt by introducing a mask adapter that uses the forensic clue to generate an automatic heatmap prompt $\mathcal{M}$ for the decoder. Possibilities for such a prompt are either a point, a bounding box, or a heatmap. We use a heatmap because it reflects the spatial structure of the forensic signal. In contrast, point- or box-based prompts largely disregard this information. The mask adapter maps all features into a heatmap prompt $\mathcal{M}$ suitable for SAM's decoder. It ingests all features $\{F_s^{\mathcal{I}}, F_s^{\mathcal{I}'_1}, \widetilde{F}_s\}$ and first bilinearly upsamples them to a common fine grid $\hat{s} = \max \mathcal{S}$. We then perform cross-scale, cross-stream convolutional fusion to obtain a unified feature tensor $F_{\text{fuse}} \in \mathbb{R}^{d \times \hat{s} \times \hat{s}}$. Such fusion is spatially consistent as in HRNet (Wang et al., 2020), and lightweight due to shallow cross-scale mixing.

To enforce global consistency, we use a lightweight transformer at a coarse resolution; its self-attention aggregates context across patch tokens and suppresses spatially inconsistent forgery estimates, see the Appendix B.3 for visual examples. Taking input $F_{\text{fuse}}$, the transformer operates on a downsampled, patchified representation to produce low-resolution coarse logits $L_{\text{coarse}}$ and an uncertainty logit map $U \in \mathbb{R}^{\hat{s} \times \hat{s}}$. Where the downsampling factor is treated as a hyperparameter. Both are upsampled back to the common grid $\hat{s}$, yielding $L_{\text{coarse}} \in \mathbb{R}^{\hat{s} \times \hat{s}}$ and $U \in \mathbb{R}^{\hat{s} \times \hat{s}}$.

Restoration of fine boundaries requires merging high-level context with local detail; we do this by mixing context through linear spatial gating, as in (Chen et al., 2016). We produce refined logits $L_{\text{refine}} \in \mathbb{R}^{\hat{s} \times \hat{s}}$ from $F_{\text{fuse}}$ via a 2-layer convolutional network. Finally, we apply a spatial gate $g \in [0, 1]^{\hat{s} \times \hat{s}}$ to linearly blend refined and coarse predictions into the decoder mask:

$$\mathcal{M} = g \, L_{\text{refine}} + (1-g) \, L_{\text{coarse}}.$$

The gate $g$ is a $1 \times 1$ convolution layer followed by a sigmoid with input $[L_{\text{coarse}}, U]$ that down-weights refinement where the coarse mask is confident (or uncertain), stopping over-sharpening in unedited regions while allowing detailed corrections where needed.

**Mask decoder** Before decoding, we bilinearly upsample the heatmap prompt $\mathcal{M}$ and adapted features $\{\widetilde{F}_s\}$ to $256 \times 256$ for finer mask generation and input them to the frozen SAM2 mask decoder to obtain forgery logits $\hat{\mathcal{M}}$ at $256 \times 256$. We choose $256 \times 256$ because it is close to the minimum image resolution in our data, which helps avoid extreme extrapolation artifacts in the final binary mask. Finally, following SAM2 precisely, we bilinearly upsample $\hat{\mathcal{M}}$ to the image resolution and convert it to a probability map via a sigmoid operation: $\sigma(\hat{\mathcal{M}})$. The final forgery binary mask is $\mathcal{B} = \mathbb{1}\{\sigma(\hat{\mathcal{M}}) \geq \frac{1}{2}\}$.

**Loss function** Training the mask and feature adapters follows SAM2's objectives (Chen et al., 2024), combining focal loss (Lin et al., 2018), Dice loss and IoU loss. Dice loss maximizes the overlap between the predicted and ground-truth masks by penalizing their normalized differences. Focal loss further addresses the class imbalance in IFL. The IoU loss trains SAM2's IoU prediction head via an $L_1$ loss on the forgery mask IoU. All losses take the ground truth and the model's predicted masks as inputs. The predicted mask is computed using only the tampered image. Formally, our final objective is

$$\mathcal{L} = \mathcal{L}_{\text{Dice}} + \lambda_{\text{focal}} \, \mathcal{L}_{\text{focal}}^{\alpha, \gamma} + \lambda_{\text{IoU}} \, \mathcal{L}_{\text{IoU}}.$$

The focusing parameter $\gamma \geq 0$ down-weights well-classified examples. The balance factor $\alpha \in [0, 1]$ re-weights positive vs negative examples to counteract class imbalance. We borrow $\lambda_{\text{focal}} = 20$, $\lambda_{\text{IoU}} = 1$ from the SAM2 paper (Ravi et al., 2024) and sweep over $(\alpha, \gamma)$.

### 3.3 AUTOEDITFORGE: FUELING DETECTIVE SAM BY AUTOMATING AI-DRIVEN EDITS

To address the critical shortage of up-to-date, high-quality testing and fine-tuning data for forgery localization models, we introduce `AutoEditForge`, a novel automated infrastructure for up-to-date periodic IFL robustness. This fully automated pipeline generates realistic image edits with pixel-accurate segmentation masks. Unlike existing synthetic datasets constrained by either labor-intensive manual annotation that limit scale or automated approaches that compromise realism through simplistic inpainting with limited edit variety (Kwon et al., 2024), `AutoEditForge` leverages state-of-the-art (SOTA) diffusion models to mimic the diversity of human-like edits, enabling continual evaluation and fine-tuning. `AutoEditForge` implements a two-pass architecture that separates lightweight analysis from computationally intensive editing operations, enabling efficient processing of large-scale image batches.

**First pass: analysis and decision making.** The first pass performs comprehensive scene analysis to identify editing opportunities. Florence-2 (Xiao et al., 2023) conducts dense image captioning and object detection with bounding box extraction. An LLM (Gemma 3 12B-it (Team et al., 2025)) then analyzes the detected objects and scene context to determine the most appropriate editing strategies for each image. The system selects from four editing methods:

- **Replace:** Substitutes existing objects with semantically similar alternatives while maintaining scene coherence. For example, replacing a golden retriever with a Labrador, or a red apple with a green pear, preserving logical consistency while introducing variation.
- **Remove:** Eliminates objects from the scene. For instance, removing a newspaper from a person reading on a park bench, filling the area utilizing contextual understanding.
- **Add:** Introduces new objects in suitable locations based on spatial and semantic analysis. Examples include adding birds to sky regions or picnic baskets to grass areas, respecting scene perspective and environmental coherence.
- **Change Partially:** Alters object attributes while preserving the object's identity and overall structure. This enables transformations such as material changes (wooden to metal chair), texture modifications (plain to brick wall), or style updates (modern to vintage car design).

**Second pass: segmentation and inpainting.** The second pass executes the specific editing operations determined in the first pass. SAM2 (Ravi et al., 2024) generates precise pixel-level segmentation masks using bounding box coordinates from Florence-2's object detection. Instruction-based diffusion image editing models then perform the actual image editing operations based on the selected strategy and target regions. The pipeline includes several post-processing techniques to ensure robustness: hole filling for mask continuity, disconnected component analysis for fragmented objects, size-based filtering to remove spurious detections, and morphological operations for mask refinement. For implementation details and prompting examples, see Appendix D.

**Detective SAM and AutoEditForge.** `AutoEditForge` supplies a steady stream of realistic, instruction-guided edits from the latest generative editing models. `Detective SAM` ingests this stream via adapter fine-tuning, which aligns the frozen SAM2 decoder and its prompts to the current distribution of editing techniques. The result is a practical lifelong learning loop: evaluate on fresh edits, surface errors, fine-tune adapters, and redeploy, all while keeping the backbone fixed and maintaining robustness across evolving editors and instructions.

## 4 EXPERIMENTS

**Training specification.** Detective SAM is trained on 10k samples of SIDA (Huang et al., 2025) and all 8807 train samples of MagicBrush (Zhang et al., 2024). We OOD test on CoCoGLIDE, UltraEdit (Zhao et al., 2024), AutoSplice (Jia et al., 2023), NanoBanana (Comanici et al., 2025); NanoBanana is generated with AutoEditForge. All datasets are diffusion-edited; full details in Appendix G. Detective SAM$^{SOTA}$ is fine-tuned on 500 samples of FLUX-Bench (Labs et al., 2025) and QWEN-Bench (Wu et al., 2025) (1000 total, created with AutoEditForge). Therefore, CoCoGLIDE, AutoSplice and NanoBanana are always entirely OOD. The noise intensity is tuned over a range of six values, where the values depend on the noise type. Other hyperparameters are tuned over a grid as in the Appendix H.

**Testing setup.** Our results are divided into three regimes: (1) **In-Distribution (ID)**: Test on the out-of-sample test set of our training set. (2) **Out-Of-Distribution (OOD)**: Test on completely unseen test sets for a fair comparison to baselines. (3) **Fine-tuned**: The pretrained Detective SAM is fine-tuned on 500 samples of the respective datasets to evaluate adaptation efficiency. **Fine-tuning.** Fine-tuning of Detective SAM is performed with the concept of direct replay (Zhou et al., 2024). We mix 20% of the original MagicBrush & SIDA training data with our new AutoEditForge samples to mitigate catastrophic forgetting. The loss function remains unchanged, and validation is done on the relative validation mix of replay and fine-tune data.

**Evaluation Metrics.** Performance is evaluated with pixel-level mean Intersection over Union (IoU) and mean F1 score. IoU measures the overlap between the ground truth forged mask and $\mathcal{B}$, and F1 score serves as harmonic mean between pixel-level precision and recall. See also Appendix E.1.

**Baselines.** Detective SAM's forgery localization performance is evaluated against a comprehensive list of recent baseline models: SAFIRE (Kwon et al., 2024), Mesorch (Zhu et al., 2024b), TruFor (Guillaro et al., 2023), AdaIFL (Li et al., 2025), PSCC-Net (Liu et al., 2022) and the MLLM localizers SIDA-7B (Huang et al., 2025) and FakeShield (Xu et al., 2025). The total parameter count and computation per inference differ significantly. SIDA has 7B parameters, FakeShield has 23B, and SAFIRE uses 256 parallel SAM inferences for each sample. All inference is done on a single NVIDIA H100 GPU; see Appendix F.6 for the throughput of each model. Performance is judged purely on OOD scores for a fair comparison.

## 4.1 RESULTS

We present our results in two parts. **First**, we showcase Detective SAM's and the baselines' performance on OOD data. **Second**, we showcase the results on our harder AutoEditForge state-of-the-art datasets to highlight performance collapse and Detective SAM's efficient fine-tuning.

**Comparison with state-of-the-art (SOTA) methods.** Table 1 compares the baseline against Detective SAM's performance. On the four OOD datasets (CoCoGLIDE, UltraEdit, AutoSplice, NanoBanana), Detective SAM significantly outperforms the baselines. We notice strong results of several baselines on particular datasets; e.g., SAFIRE scores an F1 score of 46.38 on CoCoGLIDE, but the performance significantly degrades on all other datasets. Hence, we also present the average IoU and F1 across the four OOD datasets. Table 1 shows that TruFor is the strongest average baseline. All models suffer a significant performance drop on NanoBanana, our most recent diffusion model dataset. Only two rows in Table 1 are ID, while the rest are OOD, which reflects the intended operating regime, being more diagnostic of real-world reliability.

We underscore Detective SAM's generalization performance. Whereas most models have unstable scores over datasets, Detective SAM has similar in- and out-of-distribution scores and has the highest OOD scores (IoU = 34.68 and F1 = 42.03). Note that TruFor and SAFIRE report an alternative F1 score calculation; for more information on comparability, see Appendix E.2.

Table 1: Six-benchmark evaluation. **Legend:** ID ▨, OOD ▨. All baselines are run inference-only with appropriate preprocessing. The last column contains the average scores for CoCoGLIDE, AutoSplice, and NanoBanana (OOD for all models).

| Model | *MagicBrush* | | *SIDA* | | CoCoGLIDE | | UltraEdit | | AutoSplice [1] | | NanoBanana | | Avg OOD | |
|---|---|---|---|---|---|---|---|---|---|---|---|---|---|---|
| | IoU ↑ | F1 ↑ | IoU ↑ | F1 ↑ | IoU ↑ | F1 ↑ | IoU ↑ | F1 ↑ | IoU ↑ | F1 ↑ | IoU ↑ | F1 ↑ | IoU ↑ | F1 ↑ |
| SAFIRE [2024] | 21.02 | 27.04 | 21.35 | 27.43 | 42.22 | 46.38 | 18.41 | 24.00 | 18.71 | 24.53 | 11.39 | 15.25 | 22.68 | 27.54 |
| Mesorch [2024b] | 16.18 | 27.36 | 13.19 | 20.29 | 36.45 | 44.50 | 5.45 | 7.51 | 27.53 | 38.72 | 10.22 | 13.85 | 19.91 | 26.15 |
| TruFor [2023] | 26.41 | 34.55 | 20.08 | 28.35 | 37.76 | 45.82 | 16.15 | 22.35 | 43.34 | 58.87 | 2.59 | 3.19 | 24.96 | 32.55 |
| AdaIFL [2025] | 12.18 | 20.99 | 12.77 | 18.98 | 20.90 | 26.58 | 7.73 | 11.23 | 11.23 | 33.73 | 8.70 | 11.95 | 12.14 | 20.87 |
| SIDA [2025] | 22.94 | 26.57 | 39.12 | 52.87 | 13.24 | 15.53 | 3.29 | 4.45 | 39.31 | 48.28 | 0.09 | 0.02 | 13.98 | 17.07 |
| FakeShield [2025] | 8.81 | 12.08 | 11.66 | 13.77 | 13.72 | 14.99 | 12.98 | 18.32 | 23.75 | 29.53 | 9.57 | 10.75 | 15.01 | 18.40 |
| PSCC-Net [2022] | 10.15 | 9.80 | 2.50 | 3.49 | 31.55 | 37.60 | 10.06 | 15.43 | 36.68 | 42.43 | 12.73 | 13.26 | 22.76 | 27.18 |
| Detective SAM | **46.48** | **57.55** | **54.55** | **65.29** | **44.74** | **51.50** | **27.74** | **35.54** | **46.90** | **60.30** | **19.34** | **20.77** | **34.68** | **42.03** |

**Visual results.** Figure 3 showcases the mask predictions for each baseline and Detective SAM. We observe inconsistent results over the datasets, with multiple models detecting SOTA images as authentic (black mask) while correctly localizing legacy (AutoSplice, CoCoGLIDE) samples. We provide several low IoU Detective SAM failure cases for each dataset in the Appendix F.5.

---

[1] AutoSplice shares the same editing model as MagicBrush, see Appendix G.

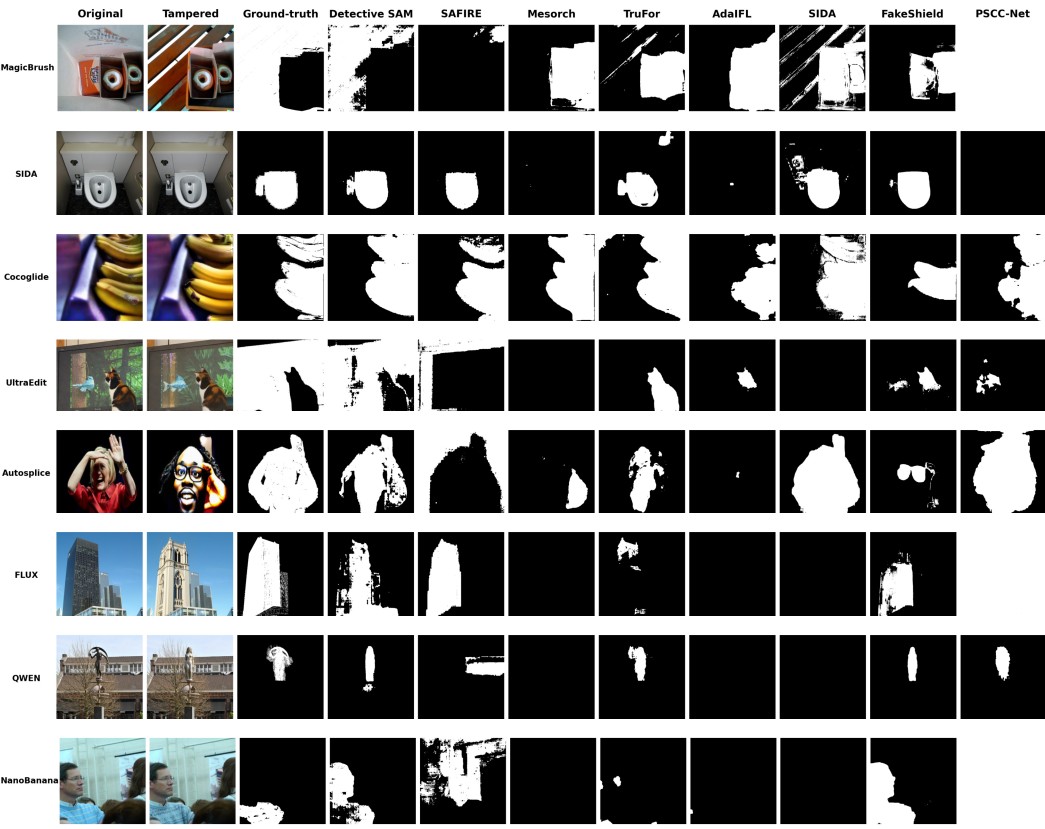

Figure 3: Overview of qualitative results across all models and datasets. Each row corresponds to a dataset sample and each column to the original and tampered images, the ground-truth mask, and each model's predicted mask. For SIDA, the original and tampered are equal, since (original, tampered) pairs are not provided in the test set.

**Model collapse and fine-tuning.** We investigate the performance on SOTA AutoEditForge datasets and analyze Detective SAM's lightweight fine-tuning. Table 2 shows the scores for our created SOTA datasets: FLUX-Bench, QWEN-Bench and NanoBanana. Focusing on all models, we notice an all-round performance drop. SAFIRE outperforms on QWEN-Bench, with Detective SAM showing stable results across all SOTA datasets [2].

Although outperformance on prevailing benchmarks is often taken as evidence of generalization in IFL, the results demonstrate that such gains do not carry over to SOTA diffusion-based edits, as none of the evaluated detectors generalize effectively and all exhibit substantial degradation. This emphasizes the need for periodic adaptation to future-proof systems, e.g., periodic fine-tuning, as increasingly more capable models are released.

As SAM2's backbone weights are frozen, and our adapters are lightweight, Detective SAM lends itself to efficient fine-tuning. We fine-tune Detective SAM on 500 samples of both FLUX-Bench and QWEN-Bench (not NanoBanana), to create Detective SAM$^{SOTA}$, shown in the final row of Table 2. Fine-tuning restores Detective SAM's capabilities on both FLUX-Bench and QWEN-Bench datasets, with an IoU of 43.08 and 41.44, respectively. Consider that these datasets are now ID for Detective SAM$^{SOTA}$, and therefore cannot be compared to baselines' results directly in Table 2. Detective SAM$^{SOTA}$'s average OOD performance improves to an IoU of 35.57 and F1 of 45.62. This can be attributed to significantly increased performance on NanoBanana due to the exposure to the more recent FLUX and QWEN data. Full scores are in the Appendix F.3.1.

---

[2]FakeShield (Xu et al., 2025) underperforms on diffusion edits, consistent with its reported AIGC results. However, they report strong results on traditional copy-move and splicing forgery. SIDA's (Huang et al., 2025) low score is due to the detect-then-localize pipeline misidentifying tampered images as authentic.

Table 2: FLUX-Bench, QWEN-Bench, and NanoBanana results for all baselines, Detective SAM and the fine-tuned Detective SAM$^{SOTA}$. **Legend:** ID ▨, OOD ▢. Gray rows were used to fine-tune Detective SAM (ID); others are OOD.

| Model | FLUX-Bench | | QWEN-Bench | | NanoBanana | |
|---|---|---|---|---|---|---|
| | IoU ↑ | F1 ↑ | IoU ↑ | F1 ↑ | IoU ↑ | F1 ↑ |
| SAFIRE [2024] | 19.03 | 22.53 | **21.72** | **26.87** | 11.39 | 15.25 |
| Mesorch [2024b] | 10.59 | 14.35 | 10.04 | 15.26 | 10.22 | 13.85 |
| TruFor [2023] | **19.34** | **22.67** | 19.42 | 22.76 | 2.59 | 3.19 |
| AdaIFL [2025] | 6.62 | 9.21 | 6.62 | 8.40 | 8.70 | 11.95 |
| SIDA [2025] | 0.89 | 0.99 | 0.82 | 0.95 | 0.09 | 0.02 |
| FakeShield [2025] | 8.57 | 9.51 | 9.77 | 11.04 | 9.57 | 10.75 |
| PSCC-Net [2022] | 12.65 | 14.83 | 13.46 | 15.28 | 12.73 | 13.26 |
| Detective SAM | 18.70 | 21.28 | 20.41 | 22.29 | **19.34** | **20.77** |
| Detective SAM$^{SOTA}$ | 43.08 | 52.52 | 41.44 | 51.49 | 27.00 | 36.21 |

Finally, we fine-tune Detective SAM$^{SOTA}$ on different amounts of samples, with and without replay. In the Appendix F.3.4, Figure 13a, we observe an initial drop in OOD performance due to overfitting on the limited number of samples. As the number of samples increases, the ID and OOD performance is retained while increasing the fine-tuned scores. The importance of replay is evident from the ID performance drop and lower OOD scores in Figure 13b.

**Impact of edit method.** Appendix F.4 reports average IoU by edit method. The largest gap is between Replace and Remove: on QWEN-Bench, 22.95 vs 10.58, a 116.92% difference; on FLUX-Bench, 17.61 vs 9.31, an 89.15% difference. This suggests that SOTA datasets require editing methods that are more diverse than just inpainting; otherwise, methods such as removal will not be detected in-the-wild.

**Impact of external perturbations.** Appendix F.2 shows the effect of pre-processing the images with increasing perturbations for Gaussian Blur, Gaussian Noise, and JPEG compression. Upon analysing the figures, we observe that Detective SAM is relatively robust to Gaussian Blur and Noise. These are the types of perturbations that are used as forensic clues. This suggests that external noise added to the input affects both streams similarly, and the difference between them remains informative. In terms of robustness to JPEG compression, Detective SAM is in line with the baselines.

## 4.2 ABLATION STUDIES

All ablations are run on our training set of SIDA and MagicBrush and their validation splits with SAM2 frozen and identical training/tuning protocol.

**Impact of perturbation type.** Table 3a shows that the perturbation type has a significant impact on the localization performance. We notice that Gaussian blur performs well, and that a combination of Gaussian noise and blur performs marginally better. This is in line with MINDER (Tsai et al., 2024b), which combines both types for improved image forgery localization performance. However, adding JPEG further improves performance, demonstrating the practical strength of explicit perturbation signals. This paper uses Gaussian Blur and Noise to align with prior findings.

**Mask adapter design.** Analyzing Table 3b, our more intricate architecture (downscaled transformer, uncertainty, and spatial gating, see Section 3.2) improves validation performance compared to a straightforward convolutional network. Feature adapters drive the largest gains by enabling forensic embeddings to exploit SAM2's decoder image-prior for forgery localization.

**Impact of mask decoder.** We train Detective SAM using the heatmap from the mask adapter directly for localization without SAM2's decoder. Displayed in the final row of Table 3b. The significant performance drop without the mask decoder indicates a substantial benefit from utilizing the information contained in SAM2's decoder training.

**Impact of noise intensity.** The noise intensity for a perturbation is chosen as the value with the highest validation performance over a range of six values. The best performing intensities for Gaussian noise & blur combination is plotted in the Appendix F.1.

Table 3: Detective SAM ablation study using the validation performance on SIDA and MagicBrush.

(a) Perturbation ablation

| Perturbation | IoU ↑ | F1 ↑ |
|---|---|---|
| JPEG + Noise + Blur | 52.58 | 63.08 |
| Noise + Blur | 50.52 | 61.42 |
| JPEG + Blur | 48.66 | 59.08 |
| Gaussian Blur | 48.17 | 57.78 |
| JPEG + Noise | 46.56 | 56.95 |
| Noise | 43.44 | 52.60 |
| JPEG | 42.56 | 51.02 |
| None | 36.22 | 44.75 |

(b) Architectural ablation

| Configuration* | IoU ↑ | F1 ↑ |
|---|---|---|
| Detective SAM | 50.52 | 61.42 |
|   Simple convolution | 44.48 | 54.99 |
|    w/o Feature adapters | 14.29 | 20.21 |
| Without decoder | 36.41 | 47.81 |

* indentation implies cumulative ablation.

## 5 CONCLUSION

`Detective SAM` advances diffusion-based forgery localization, reaching a mean out-of-distribution IoU of 34.68, representing a 38.94 % increase across out-of-distribution baselines and over four test-sets. It has been demonstrated that IFL systems exhibit superior performance in the presence of strong, explicit perturbation-based forensic signals that incorporate a robust segmentation backbone. Furthermore, the efficacy of up-to-date periodic fine-tuning has been established as a prerequisite for the advent of novel diffusion editors, a process that AutoEditForge facilitates.

**Limitations.** Our reliance on perturbation-driven cues makes performance sensitive to both the specific cues and the strength of the perturbation. Further research should investigate adaptive perturbations and increasing the number of perturbations, as validation performance seems to increase with the number of perturbations. Classical copy-move and splicing forgeries do not contain the same diffusion-sensitive artifacts; thus, different signals and broader training are required and should be investigated. A ready-to-deploy model should use training on fully synthetic and authentic images to mitigate false positives/negatives.

By articulating these steps, we aim to advance the IFL field further to keep pace with the evolving generative editing tools.

**Reproducibility Statement.** To ensure the reproducibility of our research, we open-source the code for AutoEditForge, Detective SAM training, and the pretrained weights at the anonymized repository `https://anonymous.4open.science/r/Detective-SAM-9057/`. The NanoBanana, QWEN-Bench, and FLUX-Bench datasets will be released upon acceptance. The model is trainable on a single NVIDIA H100 GPU. Other datasets in this paper (MagicBrush Zhang et al. (2024), SIDA Huang et al. (2025), AutoSplice Jia et al. (2023), CoCoGLIDE) and baselines (SAFIRE Kwon et al. (2024), Mesorch Zhu et al. (2024b), TruFor Guillaro et al. (2023), AdaIFL Li et al. (2025), FakeShield Xu et al. (2025), PSCC-Net Liu et al. (2022)) are publicly available.

**Ethics Statement.** Detective SAM is designed for the forensic localization of diffusion-based edits to support provenance research and platform integrity, and its outputs should be treated as probabilistic evidence, subject to human oversight. The system is dual-use; adversaries may exploit failure modes, or misinterpretations may harm stakeholders. Therefore, we recommend per-model validation with AutoEditforge and human-in-the-loop review. We train and evaluate on public datasets and edits from AutoEditForge; no new personal data is collected, and we will honor take-down requests. On AI usage, Large Language Models were used for writing assistance and code completion; all ideas and analyses are our own.

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

# A    NOTATION

Table 4: Notation used in `Detective SAM` and `AutoEditForge`.

| Symbol | Meaning | Type or shape |
|---|---|---|
| $\mathcal{I}$ | RGB source image | $\mathbb{R}^{3 \times H \times W}$ |
| $H, W$ | Image height and width | $\mathbb{N}$ |
| $\mathcal{B}$ | Binary forgery mask | $\{0, 1\}^{H \times W}$ |
| $N$ | Number of perturbations | $\mathbb{N}$ |
| $\text{Perturb}_i(\cdot; \theta)$ | Image perturbation operator $i$ with params $\theta$ | Function |
| $\theta$ | Perturbation parameters | Hyperparameters |
| $\mathcal{I}'_i$ | Perturbed image $i$ | $\mathbb{R}^{3 \times H \times W}$ |
| $\mathcal{S}$ | Set of HIERA scales | $\{32, 64, 128\}$ |
| $X_s$ | HIERA embedding at scale $s$ | $\mathbb{R}^{C_s \times s \times s}$ |
| $\text{ConvSAM}(\cdot)$ | Frozen SAM2 conv processing | $X_s \mapsto F_s$ |
| $C_s$ | Channels of HIERA embedding at scale $s$ | $\mathbb{N}$ |
| $F_s$ | Processed SAM2 feature at scale $s$ | $\mathbb{R}^{C_s \times s \times s}$ |
| $F_s^{\mathcal{I}}$ | Feature of $\mathcal{I}$ at scale $s$ | $\mathbb{R}^{C_s \times s \times s}$ |
| $F_s^{\mathcal{I}_i}$ | Feature of i'th perturbed image at scale $s$ | $\mathbb{R}^{C_s \times s \times s}$ |
| $\mathcal{A}_s$ | Feature adapter at scale $s$ (1×1 conv) | $[F_s^{\mathcal{I}}, F_s^{\mathcal{I}'_1}] \mapsto \Delta F_s$ |
| $\Delta F_s$ | Residual correction from feature adapter | $\mathbb{R}^{C_s \times s \times s}$ |
| $\widetilde{F}_s$ | Adapted feature $F_s^{\mathcal{I}} + \Delta F_s$ | $\mathbb{R}^{C_s \times s \times s}$ |
| $\hat{s}$ | Finest grid resolution used by mask adapter | $\max \mathcal{S}$ |
| $F_{\text{fuse}}$ | Cross-scale fused feature tensor | $\mathbb{R}^{d \times \hat{s} \times \hat{s}}$ |
| $d$ | Channel dimension of $F_{\text{fuse}}$ | $\mathbb{N}$ |
| $\mathcal{M}$ | Heatmap prompt logits for decoder | $\mathbb{R}^{\hat{s} \times \hat{s}}$ |
| $L_{\text{coarse}}$ | Coarse logits from transformer block | $\mathbb{R}^{\hat{s} \times \hat{s}}$ |
| $U$ | Uncertainty logit map | $\mathbb{R}^{\hat{s} \times \hat{s}}$ |
| $L_{\text{refine}}$ | Refined logits from conv block | $\mathbb{R}^{\hat{s} \times \hat{s}}$ |
| $g$ | Spatial gate | $[0, 1]^{\hat{s} \times \hat{s}}$ |
| $\hat{\mathcal{M}}$ | Decoder logits at output resolution | $\mathbb{R}^{H \times W}$ |
| $\sigma(\cdot)$ | Elementwise sigmoid | $(0, 1)$ mapping |
| $\lambda_{\text{focal}}, \lambda_{\text{IoU}}$ | Loss weights | $\mathbb{R}_{\geq 0}$ |
| $\alpha, \gamma$ | Focal loss parameters | $\alpha \in [0, 1], \ \gamma \geq 0$ |
| $\mathcal{L}_{\text{Dice}}$ | Dice loss | Scalar |
| $\mathcal{L}_{\text{focal}}^{\alpha, \gamma}$ | Focal loss | Scalar |
| $\mathcal{L}_{\text{IoU}}$ | IoU $L_1$ regression loss for SAM2 head | Scalar |

# B    VISUALIZATIONS

## B.1    NANOBANANA INTRODUCTION VISUALIZATION.

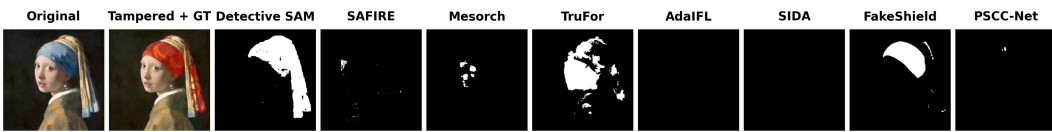

Figure 4: Source, tampered & ground-truth, mask prediction results for all baselines and Detective SAM$^{SOTA}$ for a NanoBanana example.

## B.2 SAM2 ARCHITECTURE.

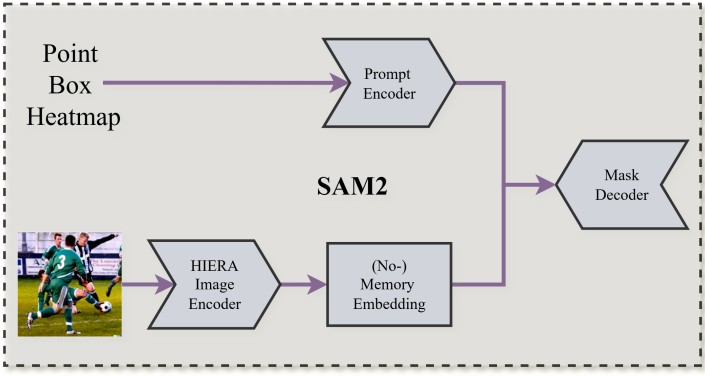

Figure 5: Original SAM2 architectural interactions for the components in Figure 2. This is an image-only version of the architecture presented in the SAM2 paper (Ravi et al., 2024).

## B.3 COARSE AND FINE MASK ADAPTER LOGITS.

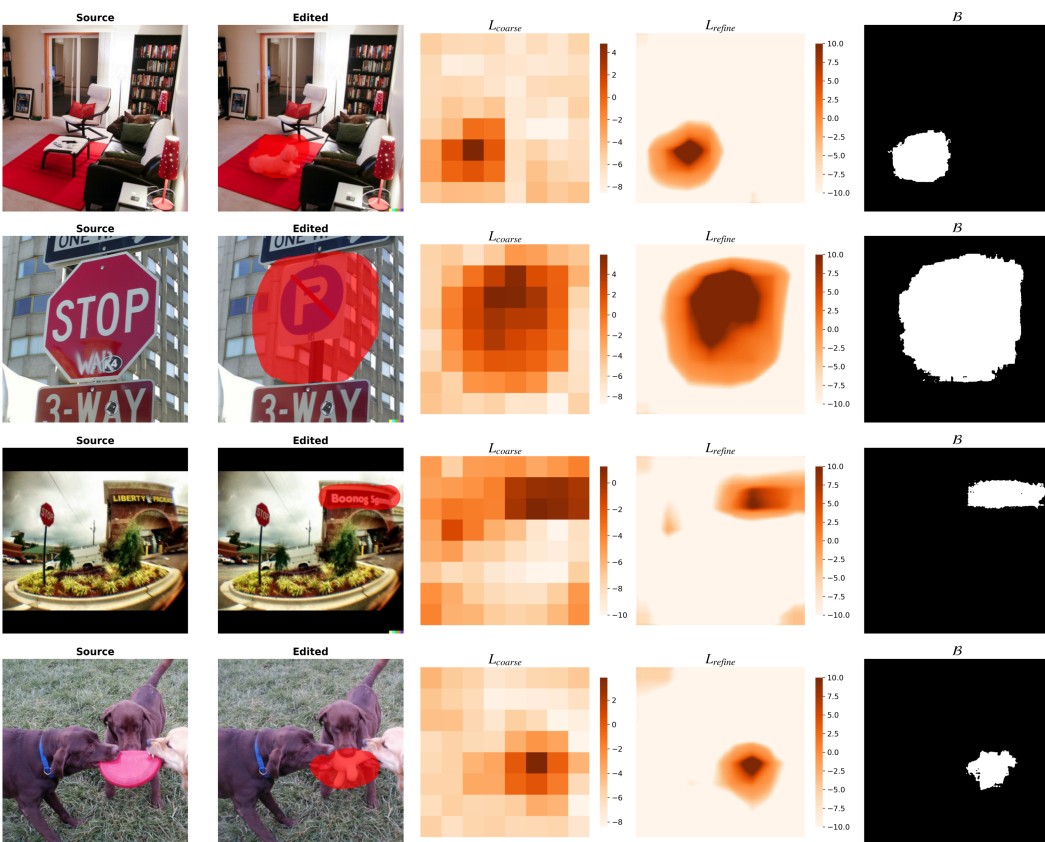

Figure 6: Examples showing mask adapter outputs on four MagicBrush training samples. For each sample we show the coarse logits $L_{\text{coarse}}$, the refined logits $L_{\text{refine}}$, and the final binary mask $\mathcal{B}$. Heatmaps are logits before sigmoid; $\mathcal{B}$ is obtained by thresholding $\sigma(\hat{\mathcal{M}})$ at $\frac{1}{2}$.

## B.4 Δ CORRECTION FEATURES

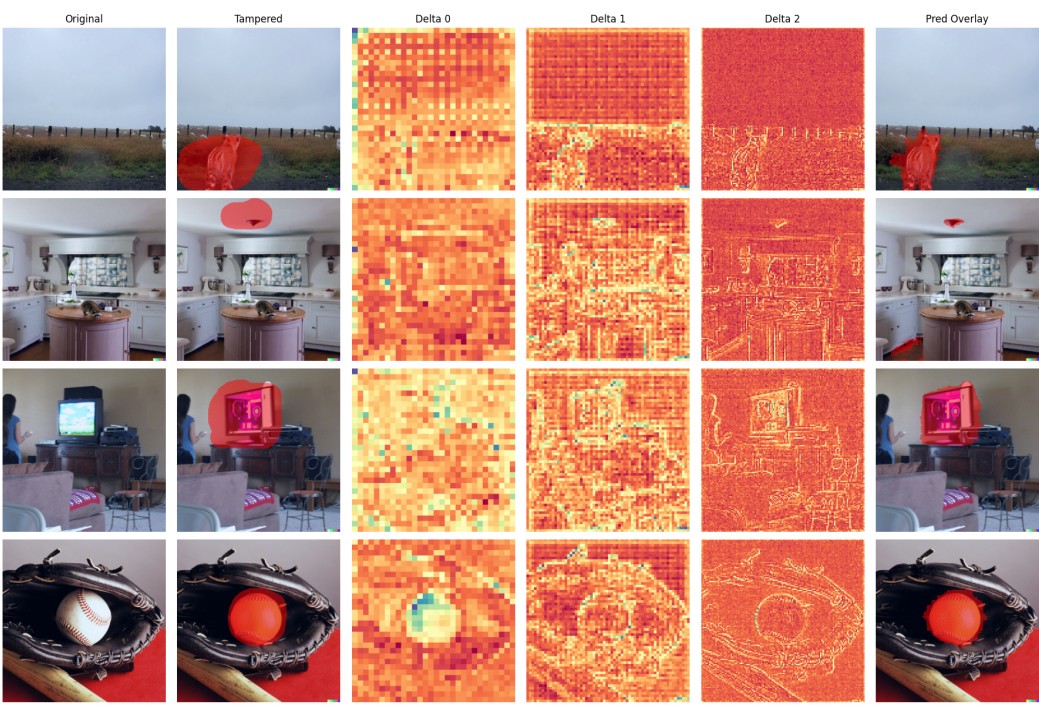

(a) Delta correction saliency results for MagicBrush.

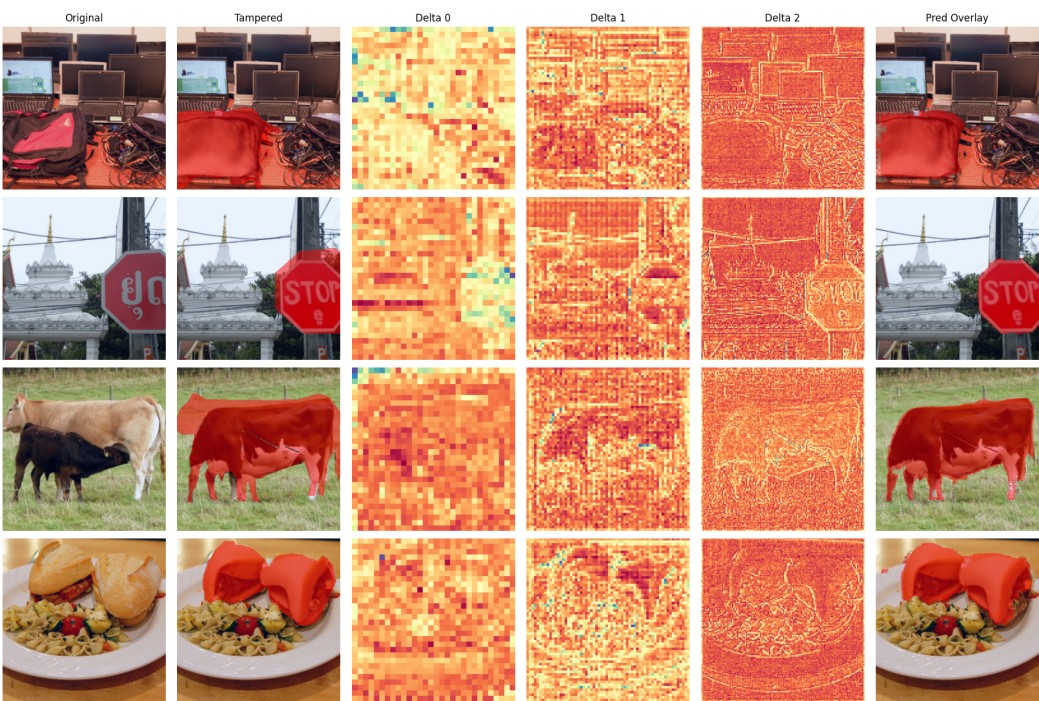

(b) Delta correction saliency results for CoCoGLIDE.

Figure 7: Delta correction saliency visualizations across MagicBrush and CoCoGLIDE for four samples, averaged over the embedding dimension and bilinearly upsampled to 512 X 512.

## C  DATA GENERATION COMPARISON

Table 5: Comparison of diffusion-based image editing data generation approaches. public availability reflects the state as of *Sep 24, 2025*.

| Dataset | Methods Used | Model Type | Public |
|---|---|---|---|
| RADAR Costanzino et al. (2025) | replace | text-conditioned inpainting | ✗* |
| GRE Sun et al. (2023) | add; remove; replace | text-conditioned inpainting | ✗ |
| SAFIRE Kwon et al. (2024) | replace; remove | text-conditioned inpainting | ✓ |
| SID-Set Huang et al. (2025) | change partially; replace | text-conditioned inpainting | ✓ |
| **AutoEditForge** Ours | add; change partially; remove; replace | instruction-based editing | ✓ |

*Dataset announced but not yet publicly released.

**Model type distinction** The datasets in Table 5 employ two fundamentally different editing approaches. Text-conditioned inpainting (used by RADAR, GRE, SAFIRE, and SID-Set) requires complete textual descriptions of desired content in masked regions, treating them as areas to be entirely regenerated. This often results in visible boundaries and loss of contextual details like consistent lighting and perspective. In contrast, our `AutoEditForge` uses instruction-based editing models, which can interpret natural language commands (e.g. "replace the dog with a cat") to perform targeted modifications while preserving scene coherence. Although instruction-based models can operate without masks, `AutoEditForge` employs segmentation masks to ensure precise spatial control, combining semantic understanding with spatial precision for context-aware edits that maintain the original scene's lighting, perspective, and style.

**RADAR** Costanzino et al. (2025) employs a systematic pipeline that uses Kosmos-2 for scene analysis and object detection, followed by Grounded SAM for segmentation of selected objects. The system focuses on replacement operations, using the original scene caption as an inpainting prompt across 10 different text-to-image diffusion models to generate semantically coherent substitutions (e.g., replacing a duck with another bird). Unlike RADAR which focuses solely on object replacement using scene-level captions, `AutoEditForge` employs LLM-guided decision making to support diverse editing operations (add, remove, replace, change partially) with context-aware prompting.

**GRE** Sun et al. (2023) employs a comprehensive multi-stage pipeline that leverages large models across different modalities, including SAM for region selection, BLIP2 for scene understanding, and ChatGPT for generating logical editing ideas to ensure semantically coherent edits. The system performs three types of operations (add, remove, replace) using diverse editing methods spanning GAN-based (MAT, LaMa), diffusion-based (Stable Diffusion, ControlNet, PaintByExample), and black-box approaches (Photoshop with generative AI). Built on 228,650 images from real-world sources focusing on daily snapshots and news visuals, the dataset's simulated pipeline ensures logical consistency while maintaining scalability, though the dataset remains private despite its significant scale. In contrast to GRE's BLIP2-ChatGPT pipeline for text-to-image inpainting, `AutoEditForge` employs a two-pass architecture with Florence-2 and Gemma 3 12B-it for more efficient processing, extends editing capabilities with a novel 'Change Partially' operation, and supports SOTA image editing models such as Qwen-Image-Edit Wu et al. (2025).

**SAFIRE-AUTO** Kwon et al. (2024) generates a large-scale pretraining dataset of approximately 123,000 images by leveraging SAM's automatic mask generation to partition authentic images from DPReview into semantic regions, then randomly selecting and unioning adjacent regions to create manipulation masks. The pipeline applies four forgery types: copy-move, splicing, generative reconstruction using text-to-image models, and AI-based inpainting removal, with various post-processing techniques including resizing, blurring, noise addition, and color adjustments. Unlike `AutoEditForge`'s intelligent two-pass approach that uses Florence-2 and LLM analysis to make contextually-aware editing decisions based on scene understanding, SAFIRE-AUTO employs a simpler automated method that randomly selects and unions adjacent semantic regions without considering the semantic appropriateness of the edits.

**SID-Set** Huang et al. (2025) constructs a social media-focused dataset of 300,000 images through a four-stage pipeline: extracting objects from captions using GPT-4o, generating masks with Language-SAM, establishing replacement dictionaries for objects/attributes, and producing tampered images via Latent Diffusion. The system supports both object replacement (swapping entire objects like cat→dog) and attribute modification (changing properties like "happy dog"). In contrast

to `AutoEditForge`, which employs Florence-2 for object detection and an LLM for dynamic editing strategy selection across four manipulation types, SID-Set utilizes a pipeline with GPT-4o for caption-based object extraction and predefined replacement dictionaries, focusing specifically on object replacement and attribute modification for social media contexts.

# D  AUTOEDITFORGE

**Image selection and filtering.**   The images are selected from Open-Images V7 Kuznetsova et al. (2020) based on four complexity criteria to ensure meaningful forgery detection challenges: (1) containing $\geq 3$ objects with bounding boxes covering $\geq 2\%$ of the image area, (2) representing $\geq 2$ distinct object classes, (3) no single object dominating more than 60% of the frame, and (4) at least one non-person object present. This filtering strategy ensures that the generated forgeries involve realistic multi-object scenes rather than trivial single-object manipulations.

**Quality control mechanisms**   AutoEditForge implements several quality control mechanisms that are tracked during the generation process.

1. **Multi-metric duplicate detection:** Four complementary metrics (blob analysis, MAE, pHash, and SSIM) are used to validate meaningful inpainting changes and automatically reject failed images without retrying.

2. **Mask validation pipeline:** All masks undergo format validation, size matching, and area constraint checks, ensuring only high-quality masks proceed to inpainting.

3. **Error tracker:** Categorizes failures across 11 distinct error types.

We compose a table analyzing the error logs of FLUX-Bench and QWEN-Bench, totaling 6.000 samples. In total, 9.446 images were generated, with 3.443 failures, giving a failure rate of 36.45%. Each editing method has 25% of the images due to our class balancing. The failures are distributed as follows:

Table 6: AutoEditForge Failure Categories.

| Failure mode | Count | % of errors |
|---|---|---|
| Inpainting produced no result | 1730 | 50% |
| Florence mask coverage validity | 1506 | 44% |
| Florence captioning failed | 187 | 5% |
| LLM object selection failure | 17 | 0.5% |
| SAM segmentation mask failure | 1 | 0.03% |
| Fallback mask file creation errors | 1 | 0.03% |
| LLM edit method decision failures | 1 | 0.03% |

**Class balancing mechanism.**   To ensure balanced representation across editing methods, we implemented a dynamic class-balancing mechanism during generation. For each image, the LLM first analyzes the scene and selects the two most suitable editing methods from Replace, Remove, Add, Change Partially based on semantic and spatial constraints. The system then applies further class balancing by selecting the method that has been used less frequently between these two candidates, preventing any single manipulation type from dominating the dataset. This strategy resulted in an approximately uniform distribution with each method applied to $\sim$25% of the images, ensuring comprehensive coverage of forgery types for robust detector training. The final dataset comprises manipulated images with corresponding pixel-level ground truth masks, representing diverse editing operations across complex real-world scenes.

## D.1 AUTOEDITFORGE EXAMPLE.

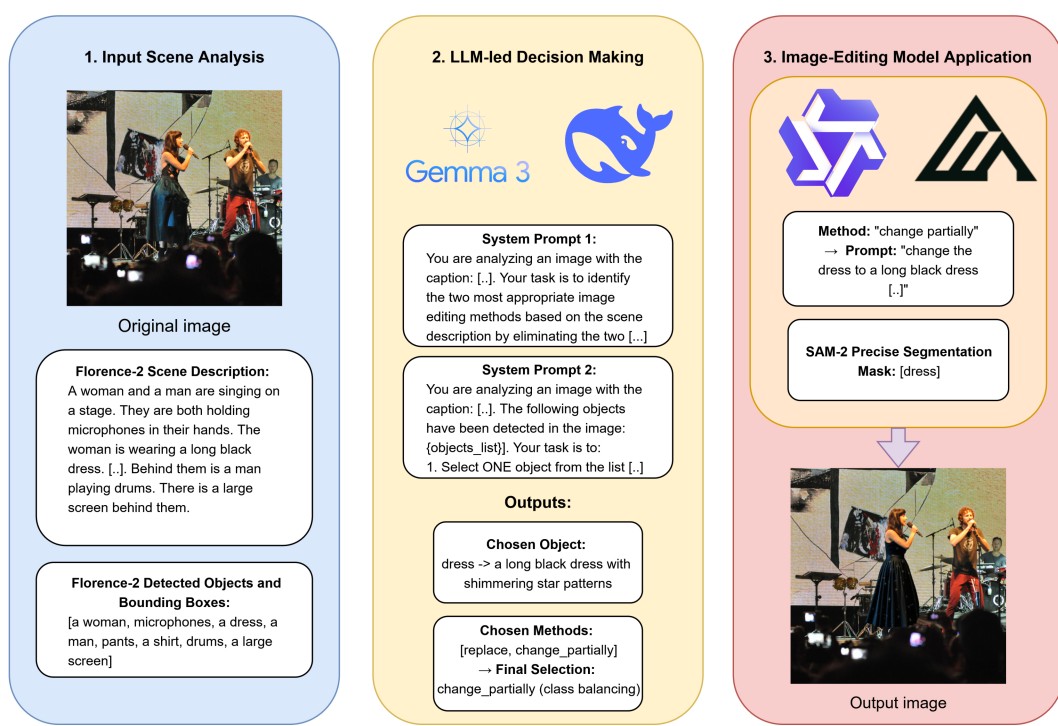

Figure 8: High-level overview of the `AutoEditForge` pipeline, illustrating the workflow from input image to edited output. Implementation details, including system prompts and source code, are available in our GitHub repository.

Further, we provide three qualitative examples of the three created datasets (FLUX-BENCH, QWEN-BENCH, NanoBanana):

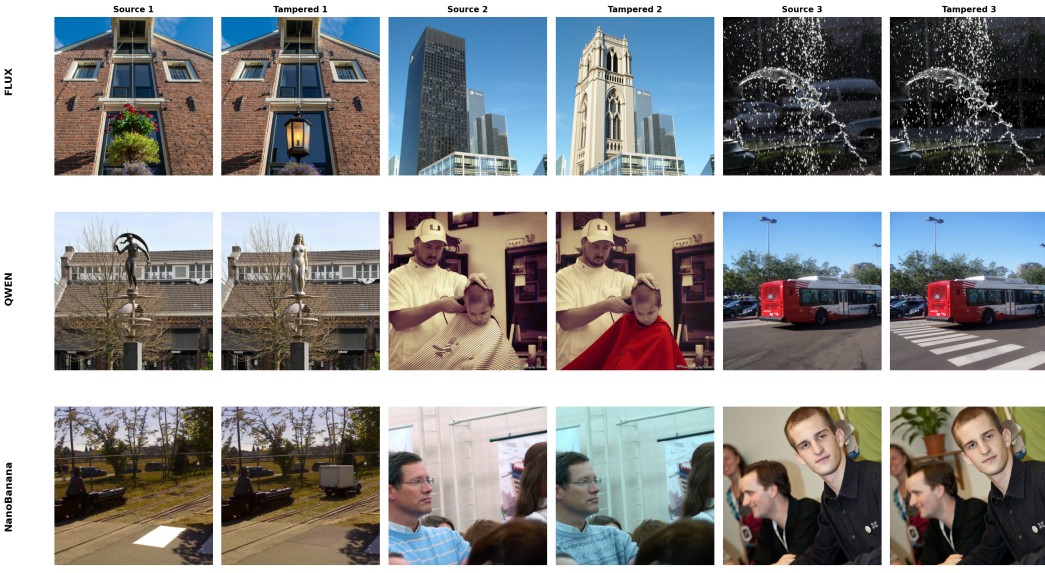

Figure 9: Qualitative comparison of three source and tampered AutoEditForge edits across FLUX-Bench, QWEN-Bench, and NanoBanana.

## D.2 DATASET CREATION TIME.

Table 7: End-to-end `AutoEditForge` generation effort. "LLM secs/img" includes edit method selection and prompt formation. "Editor secs/img" includes diffusion steps. NanoBanana edits are done via Gemini 2.5 Flash API. Comanici et al. (2025)

| Dataset | Images | LLM secs/img | Editor secs/img |
|---|---|---|---|
| FLUX-Bench | 3.000 | 35 | 41 |
| QWEN-Bench | 3.000 | 36 | 59 |
| NanoBanana | 445 | 35 | 2 |

## D.3 PERCEPTUAL QUALITY COMPARISON.

We compare the output quality of AutoEditForge using BRISQUE Mittal et al. (2012), NIQE Mittal et al. (2013), and PI Blau et al. (2019), commonly used no-reference image quality metrics for assessing perceptual differences between source and tampered images. All three aim to quantify visible degradation without relying on a pristine reference image. The metrics are non-reference, since reference metrics measure similarity, which is directly biased by the mask size. As can

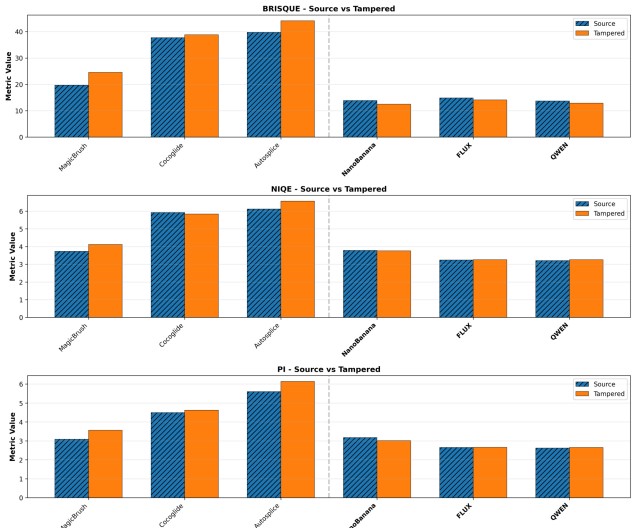

Figure 10: Source vs tampered image quality metrics for datasets with matching (source, tampered) pairs: BRISQUE, NIQE, and PI across models.

be seen in Figure 10, the differences between the source and tampered images are small for the AutoEditForge datasets (FLUX, QWEN, and NanoBanana) and CoCoGLIDE, but noticeable for Magicbrush and AutoSplice. This confirms that AutoEditForge shows no significant degradation in quality with respect to the source images.

## D.4

# E EVALUATION

## E.1 EVALUATION METRICS.

F1 is a monotone transform of IoU $J$ (F1 $= \frac{2J}{1+J}$), thus F1 $\geq$ IoU. Because the nonlinearity is applied prior to averaging, mean F1 is not recoverable from mean IoU and is more tolerant of partial overlaps and small objects. Benchmarks exhibit substantial F1–IoU discrepancies, indicative of over- or under-prediction under uncertainty (Fig. 3).

## E.2 F1 SCORE COMPARABILITY.

We compute the F1 score as $F1 = \frac{2TP}{2TP+FN+FP}$, whereas TruFor and SAFIRE use $F1 = \max\{\frac{2TP}{2TP+FN+FP}, \frac{2FN}{2FN+TP+TN}\}$, which is equal or larger. Their definition is suited for image splicing (two authentic images combined), while our definition reflects diffusion edits with a clear separation of authentic and forged regions. For comparability, we report alternative F1 scores in Table 8.

Table 8: Alternative F1 scores using the definition $\max\{\frac{2TP}{2TP+FN+FP}, \frac{2FN}{2FN+TP+TN}\}$

| Model | MagicBrush F1 ↑ | SIDA F1 ↑ | CoCoGLIDE F1 ↑ | AutoSplice F1 ↑ | NanoBanana F1 ↑ | FLUX-BENCH F1 ↑ | QWEN-BENCH F1 ↑ |
|---|---|---|---|---|---|---|---|
| SAFIRE [2024] | 39.25 | 40.14 | 59.63 | 54.88 | 30.87 | 36.00 | **44.28** |
| Mesorch [2024b] | 35.54 | 35.28 | 56.51 | 63.62 | 33.90 | 28.62 | 30.78 |
| TruFor [2023] | 43.69 | 40.52 | 49.81 | 65.60 | 24.45 | **37.38** | 38.47 |
| AdaIFL [2025] | 30.64 | 33.95 | 44.54 | 56.80 | 30.64 | 25.22 | 27.95 |
| SIDA [2025] | 29.26 | 48.43 | 38.72 | 64.64 | 22.20 | 20.26 | 22.74 |
| FakeShield [2025] | 26.81 | 33.00 | 41.78 | 59.24 | 31.03 | 27.13 | 30.94 |
| PSCC-Net [2022] | 21.03 | 24.13 | 51.72 | 54.65 | 22.67 | 27.91 | 30.27 |
| Detective SAM | **59.83** | **66.53** | **60.22** | **67.60** | **37.12** | 34.40 | 39.37 |

# F ADDITIONAL RESULTS

## F.1 NOISE INTENSITY.

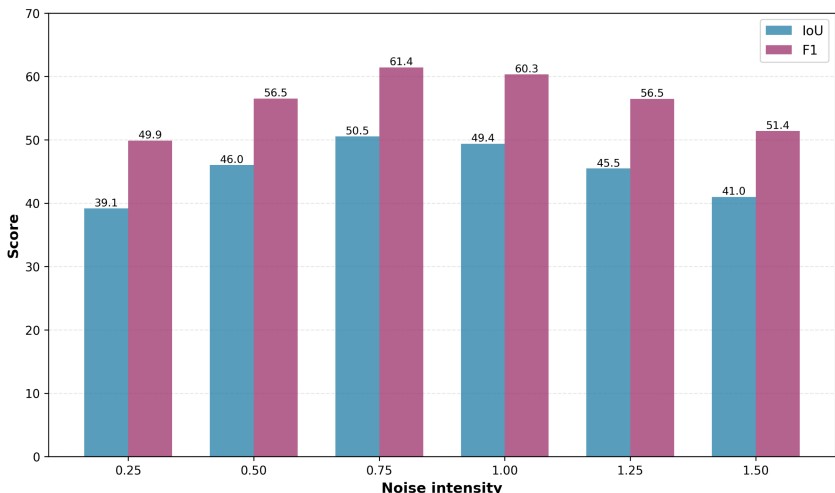

Figure 11: IoU and F1 for Gaussian noise and blur at varying intensity levels. Exact perturbation parameters for each intensity are in Table 17. Scores are averaged over the validation splits of the ID datasets (MagicBrush and SIDA).

## F.2 ROBUSTNESS STUDY.

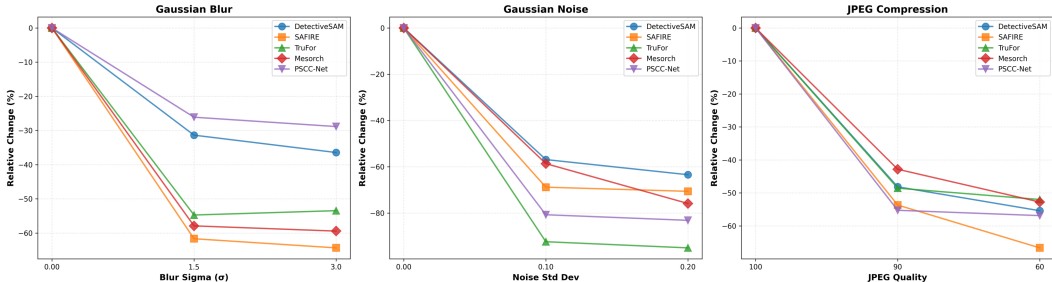

(a) Relative IoU change over Gaussian Blur, Gaussian Noise, and JPEG compression.

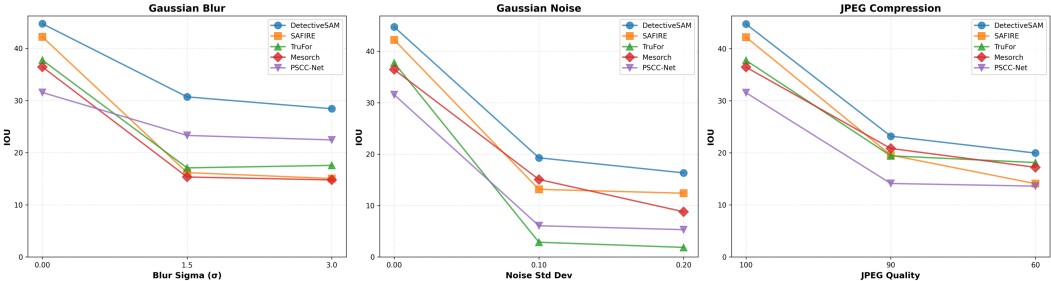

(b) IoU degradation for Gaussian Blur, Gaussian Noise, and JPEG compression.

Figure 12: Detective SAM and baselines IoU performance under increasing perturbation intensities. Only the top 5 models by IoU are shown.

## F.3 FINE-TUNING

### F.3.1 FINE-TUNING SCORES.

Table 9: Six-benchmark evaluation of the fine-tuned Detective SAM$^{\text{SOTA}}$. MagicBrush and SIDA are in-distribution (ID). CoCoGLIDE, AutoSplice, NanoBanana, and their mean form the out-of-distribution (OOD) evaluation. Bold indicates the best per column. All values are percentages; higher is better.

| Model | *MagicBrush* | | *SIDA* | | CoCoGLIDE | | UltraEdit | | AutoSplice | | NanoBanana | | Avg OOD | |
|---|---|---|---|---|---|---|---|---|---|---|---|---|---|---|
| | IoU ↑ | F1 ↑ | IoU ↑ | F1 ↑ | IoU ↑ | F1 ↑ | IoU ↑ | F1 ↑ | IoU ↑ | F1 ↑ | IoU ↑ | F1 ↑ | IoU ↑ | F1 ↑ |
| Detective SAM$^{\text{SOTA}}$ | 45.03 | 57.24 | 51.35 | 60.74 | 45.37 | 55.62 | 25.49 | 33.84 | 44.42 | 57.02 | 27.00 | 36.21 | 35.57 | 45.67 |

### F.3.2 INCREMENTAL FINE-TUNING

To further support the claim of periodic fine-tuning with Detective SAM, we fine-tune Detective SAM incrementally: first on 500 FLUX-Bench samples, then on 500 QWEN-Bench samples, and vice versa.

Table 10: Evaluate the impact of incrementally fine-tuning the IoU on the FLUX-Bench and QWEN-Bench, and vice versa. '→' denotes sequential tuning. The columns refer to the fine-tuning data used; both FLUX and QWEN refer to the use of 500 samples from the dataset. The rows refer to the dataset used to calculate the IoU.

| Dataset | Detective SAM | FLUX | QWEN | FLUX → QWEN | QWEN → FLUX |
|---|---|---|---|---|---|
| FLUX | 18.70 | 41.09 | 29.60 | 41.43 | 43.34 |
| QWEN | 20.41 | 32.26 | 42.43 | 43.20 | 42.58 |
| Average OOD [3] | 34.68 | 35.90 | 34.47 | 37.68 | 36.95 |

Examining Table 10, the first sequential update slightly reduces OOD performance, which is then restored when the following dataset is introduced. The similarity between the sequential results and

---

[3]Includes CoCoGLIDE, UltraEdit, AutoSplice and NanoBanana.

those of Detective SAM in Table 2 indicates that adaptation remains effective beyond a single update step. We will clarify the meaning of "continual" in the introduction.

### F.3.3 UNBALANCED SAMPLES FINE-TUNING

We fine-tune Detective SAM with unbalanced samples and measure IoU performance. This enables evaluation in an unbalanced setting, where one diffusion model is overrepresented during fine-tuning.

Table 11: Evaluation of unbalanced fine-tuning. The columns denote the fine-tuning data used. Only IoU results are shown. The last column is taken directly from Table 2 for reference.

| Dataset | QWEN 1500, FLUX 500 | FLUX 1500, QWEN 500 | FLUX 500, QWEN 500 |
|---|---|---|---|
| FLUX | 42.16 | 44.30 | 43.08 |
| QWEN | 44.94 | 39.68 | 41.44 |
| NanoBanana | 26.65 | 25.80 | 27.00 |
| Magicbrush | 44.14 | 45.51 | 45.03 |
| SIDA | 50.46 | 49.72 | 51.35 |
| CoCoGLIDE | 44.95 | 42.57 | 45.37 |
| AutoSplice | 43.30 | 44.31 | 44.42 |

Unbalanced fine-tuning improves performance on the overrepresented dataset but leads to greater forgetting of the underrepresented dataset. This is likely due to a mismatch between the diversity of the fine-tuning samples and the replay samples. Therefore, if future editors produce a disproportionate number of new edits, replay must either increase or subsample to maintain stability. The default setting for periodic adaptation remains 500 samples per editor with a 20% replay rate. This is supported by the results shown in Appendix F.3.4.

### F.3.4 REPLAY VS. INCREASING NUMBER OF SAMPLES.

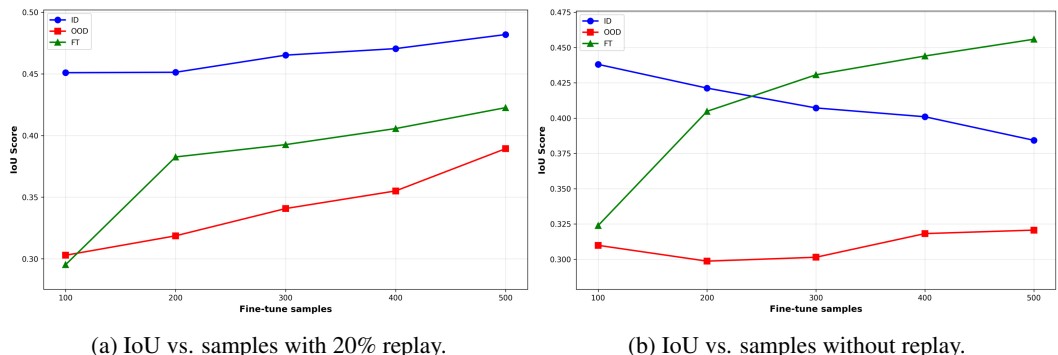

(a) IoU vs. samples with 20% replay.    (b) IoU vs. samples without replay.

Figure 13: Detective SAM fine-tuning with and without replay and increasing number of samples. The IoU average scores are shown for the ID, OOD, and fine-tuned (FT) models. Note: OOD includes CoCoGLIDE, AutoSplice and NanoBanana.

### F.4 EDIT METHODS.

Table 12: Average IoU across all evaluated models, grouped by edit operation for FLUX-Bench and QWEN-Bench. We report the mean over models.

| Dataset | Change Partially | Replace | Remove | Add |
|---|---|---|---|---|
| QWEN-Bench | 17.84 | 22.95 | 10.58 | 11.95 |
| FLUX-Bench | 17.42 | 17.61 | 9.31 | 13.27 |

### F.5 DETECTIVE SAM FAILURE MODES.

Several low IoU Detective SAM localization failures are depicted in Figure 14

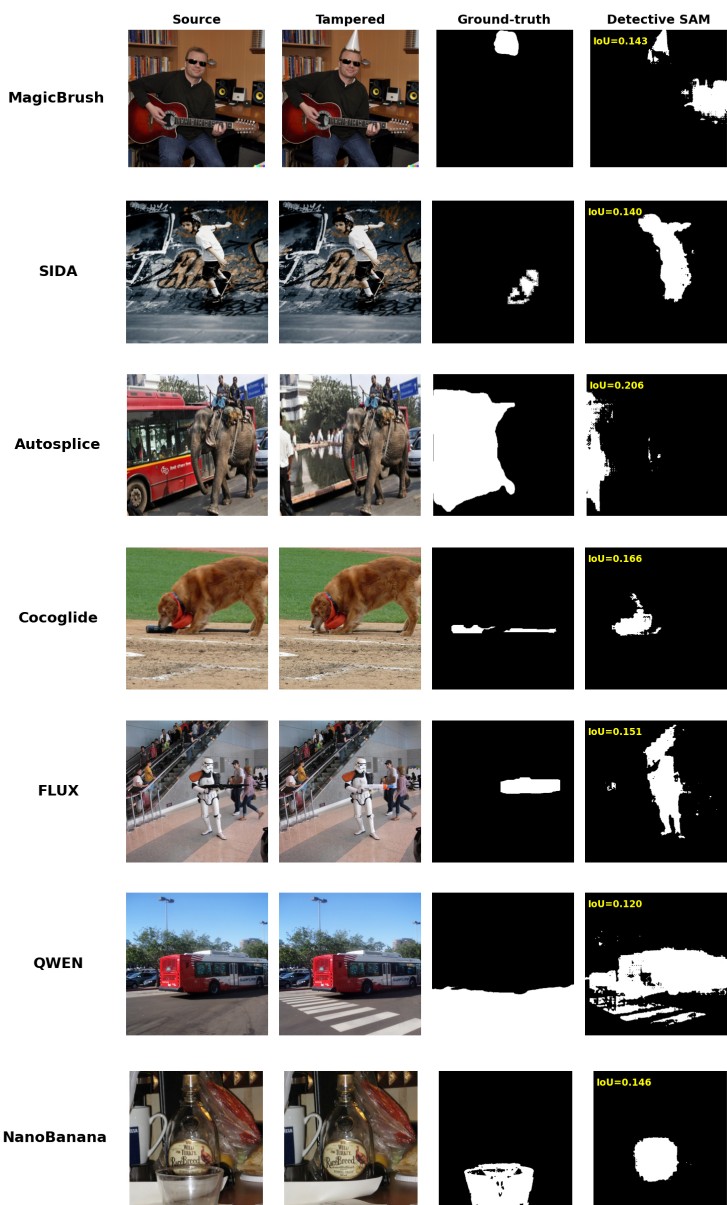

Figure 14: Detective SAM low IoU samples per dataset

## F.6 MODEL THROUGHPUT COMPARISON.

To quantify the inference efficiency difference across IFL systems, we measure throughput on 512 CoCoGLIDE samples with a batch size of 1 and no parallelization on an NVIDIA H100. The results are shown in the table below.

Table 13: Throughput and inference time comparison on CoCoGLIDE (512 samples, batch size 1, no parallel processing).

| Metric | SAFIRE | Mesorch | AdaIFL | TruFor | SIDA | FakeShield | PSCC-Net | Detective SAM |
|---|---|---|---|---|---|---|---|---|
| Images per second | 0.35 | 37.10 | 9.54 | 22.61 | 2.50 | 2.67 | **77.70** | 29.04 |
| Total inference time (s) | 1475.04 | 13.80 | 53.66 | 22.64 | 204.50 | 191.54 | **6.59** | 17.63 |
| Average OOD IoU | 24.11 | 24.73 | 27.90 | 17.43 | 17.55 | 15.68 | 26.99 | **34.68** |

From the above table, MLLM-based systems require several minutes for processing a few hundred samples, while SAFIRE requires roughly half an hour. This hinders deployment at scale. Detective

SAM achieves higher OOD performance while remaining efficient enough for large-batch screening and adaptive updates, making it suitable for practical deployment in real-time applications.

# G   DATASETS

**Datasets** We train Detective SAM on MagicBrush (Zhang et al., 2024) and a subset of SIDA (Huang et al., 2025), containing edits using DALL-E (Ramesh et al., 2022) and a Latent Diffusion Model (Rombach et al., 2022). We perform out-of-distribution testing on CoCoGLIDE, UltraEdit (Zhao et al., 2024), AutoSplice (Jia et al., 2023), and NanoBanana Comanici et al. (2025). NanoBanana is a dataset created with AutoEditForge; we also create datasets with FLUX Kontext (Labs et al., 2025) and QWEN-Image-Edit (Wu et al., 2025) to evaluate SOTA performance.
Note that MagicBrush and AutoSplice share the same editing model but differ significantly in how they create datasets. For example, the instruction, data source, editing types, and mask sizes differ. See below for all datasets and mask size details. This is in line with the OOD definition in Section 4. The editing modes for each dataset are stated in Table 14.

| Dataset | Editing Model |
|---|---|
| MagicBrush | DALL-E 2 |
| SIDA | Latent Diffusion Model |
| AutoSplice | DALL-E 2 |
| UltraEdit | SDXL-Turbo |
| CoCoGLIDE | GLIDE |
| NanoBanana | Gemini 2.5 Flash |
| FLUX-Bench | FLUX Kontext |
| QWEN-Bench | QWEN-Image-Edit |

Table 14: Overview of the editing models used for each dataset.

Next, we describe the datasets in more detail.

**MagicBrush** This dataset contains diffusion-based edits produced with DALL-E Ramesh et al. (2022); Zhang et al. (2024) using human annotation. It includes multiple edit rounds per image, and we compute binary masks as the union of forged pixels over rounds, giving 8.807 samples. We use the official validation and test split for testing, giving 528 validation samples and 1.053 test samples

**SIDA** This corpus comprises 100.000 edits created with a Latent Diffusion Model Huang et al. (2025); Rombach et al. (2022). In our experiments, we use 10.000 tampered samples of SIDA for training, 528 for validation and the full tampered test set for testing.

**AutoSplice** This dataset includes 3.621 DALL-E based edits Jia et al. (2023). We treat it as an out-of-distribution set and allocate all 3.621 images to testing. AutoSplice shares the same editing model as MagicBrush. The two datasets differ in the following aspects:

1. **Instruction:** Magicbrush contains action-oriented instructions from human crowd workers, whereas AutoSplice has descriptive captions, generated by modifying the image caption.
2. **Data source:** Magicbrush uses images from MS COCO Lin et al. (2015) and AutoSplice uses Visual News Liu et al. (2021).
3. **Edit types:** Magicbrush contains semantic changes and AutoSplice contains mainly insertions and replacements.
4. **Edit sizes:** 84% of Magicbrush edit masks cover less than 25% of the image, whereas 68% of AutoSplice masks occupy more than 25% (Table 15).

**CoCoGLIDE** This small evaluation set contains 512 GLIDE based edits Nichol et al. (2022). We use 512 samples for out-of-distribution testing.

**UltraEdit** This dataset serves as an additional OOD benchmark utilizing the SDXL-Turbo model. We use the region-based (local edited) subset, it contains 100.000 samples with pixel-level ground truth masks, from which we take a 10.000 random subset.

**NanoBanana** We construct this dataset with AutoEditForge using Gemini 2.5 Flash Comanici et al. (2025), which is not open weight and does not accept a mask input. NanoBanana generates its own internal mask during the editing process. We curate 200 images from 445 candidates and compute masks by thresholding the pixel difference between the source and edited images, selecting only the images that apply a local-only edit. All 200 samples are for out-of-distribution testing.

**FLUX-Bench** We construct this benchmark with AutoEditForge using the open weight FLUX Kontext editor Labs et al. (2025). We generate 3.000 edited samples, fine-tune on 500, validate on 250, and test on 1.750. The editor is a recent state-of-the-art model that ranks highly on public leaderboards Chiang et al. (2024).

**QWEN-Bench** We construct this benchmark with AutoEditForge using the open weight QWEN-Image-Edit model Wu et al. (2025). We generate 3.000 edited samples and fine-tune on 500, validate on 250, and test on 1.750. The editor is a recent state-of-the-art model that ranks highly on public leaderboards Chiang et al. (2024).

## G.1  DATASET MASK SIZES

Table 15: Distribution of mask sizes (small / medium / large) in each dataset. Small refers to [0,5%], medium to [5%, 25%], and large > 25% mask coverage. Percentages are rounded to integers; the last row shows the range across datasets.

| Dataset | Small (%) | Medium (%) | Large (%) |
|---|---|---|---|
| MagicBrush | 35 | 49 | 16 |
| SIDA | 32 | 47 | 21 |
| CoCoGLIDE | 20 | 45 | 36 |
| UltraEdit | 16 | 37 | 48 |
| AutoSplice | 5 | 27 | 68 |
| NanoBanana | 26 | 58 | 16 |
| FLUX-Bench | 35 | 53 | 12 |
| QWEN-Bench | 33 | 48 | 19 |

## H  HYPERPARAMETERS

Table 16: Detective SAM hyperparameters, the highest validation performance set used for the results, and the swept over range.

| Hyperparameter | Optimal | Sweep Range |
|---|---|---|
| Learning rate | 0.001 | $\{0.01, 0.001, 0.0001, 0.00001\}$ |
| Focal $\alpha$ | 0.6 | $\{0.5, 0.55, \ldots, 0.75, 0.80\}$ |
| Focal $\gamma$ | 1.0 | $\{1.0, 1.25, \ldots, 1.75, 2.0\}$ |
| Adam weight decay | 0.0001 | $\{0.0001, 0.00001, 0.0\}$ |
| Noise intensity | 0.75 | $\{0.25, 0.50, \ldots, 1.25, 1.50\}$ |
| Perturbation type | Blur & Noise | $\{$ Blur, Noise, JPEG, None, Blur & Noise $\}$ |
| Layer width | 64 | $\{$ 64, 128, 256 $\}$ |
| Transformer downscaling | $16\times$ | $\{$ 4, 8, 16 $\}$ |
| Transformer layers | 1 | $\{$ 1, 2, 3 $\}$ |
| Dropout rate | 0.15 | $\{$ 0.0, 0.1, 0.15, 0.2, 0.25, 0.5 $\}$ |
| Batch size | 4 | $\{$ 2, 4, 8 $\}$ |

Table 17: Perturbation parameters as a function of noise intensity. Each intensity level controls the strength of three perturbations: Gaussian blur with standard deviation $\sigma_{\text{blur}}$, JPEG compression with the specified quality factor (lower is stronger compression), and additive Gaussian noise with standard deviation $\sigma_{\text{noise}}$.

| Intensity | $\sigma_{\text{blur}}$ | JPEG Quality | $\sigma_{\text{noise}}$ |
|---|---|---|---|
| 0.25 | 0.25 | 80 | 0.05 |
| 0.50 | 0.50 | 66 | 0.10 |
| 0.75 | 0.75 | 52 | 0.15 |
| 1.00 | 1.00 | 38 | 0.20 |
| 1.25 | 1.25 | 24 | 0.25 |
| 1.50 | 1.50 | 10 | 0.30 |

