# OpenReview forum: "Detective SAM:  Adaptive AI-Image Forgery Localization"
_ICLR.cc/2026/Conference — ICLR 2026 Poster_

### Official Review · Reviewer_miFc · 2025-10-28

**Soundness:** 3
**Presentation:** 3
**Contribution:** 3
**Rating:** 6
**Confidence:** 4

**Summary:**

This paper presents Detective SAM, a model for localizing AI-generated image forgeries. The proposed framework builds on top of SAM2 by incorporating perturbation-driven signals to fine-tune two auxiliary modules, i.e., a mask adapter and a feature adapter, while keeping the backbone frozen. In addition to forgery localization, the authors introduce an automated pipeline that integrates instruction-based local editing methods, allowing them to evaluate the generalization ability of Detective SAM with respect to various editing techniques.

**Strengths:**

1. The paper is well-structured, and the presentation is clear and easy to follow.

2. The task of forgery localization is relevant to current research in trustworthy AI.

3. Leveraging a large foundation model like SAM2 for general-purpose forgery detection is a well-motivated and promising direction.

4. The experiments support the performance claims well. In particular, the integration of local editing methods into the evaluation pipeline adds an interesting perspective on the model’s generalization.

**Weaknesses:**

While I do not find any major flaws, I have a few questions and suggestions for improvement:

1. Perturbation design: According to Section 3.2, each image is associated with N perturbed variants (e.g., Gaussian blur, Gaussian noise, JPEG compression). Table 3(a) suggests that localization performance improves with the number of perturbations, but it is unclear whether this improvement saturates. Could the authors explore a wider range of perturbations and analyze how increasing N continues to affect performance?

2. Failure case analysis: It would be helpful to include more examples of failure cases, particularly for fine-grained or challenging forgery categories. A qualitative and/or quantitative discussion of current model limitations would enhance the reader’s understanding of where Detective SAM might still fall short.

**Questions:**

Please see the Weaknesses for details.

---

> ### Author Response · Authors · 2025-11-18
> **Response to Reviewer miFc (W1-W2)**
>
> Thank you for your valuable comments and questions. Below, we respond to the weaknesses and questions highlighted.
>
> >(W1): Could the authors explore a wider range of perturbations and analyze how increasing the number of perturbation continues to affect performance?
>
> [AW1]: To answer your question directly, we perform an additional experiment on all perturbation combinations to extend Table 3(a) in the paper:
>
> **Table C: Extending Detective SAM's Forensic Perturbations**
> | Perturbation          | IoU ↑ | F1 ↑  |
> |-|-|-|
> | JPEG + Noise + Blur   | 52.58 | 63.08 |
> | Noise + Blur          | 50.52 | 61.42 |
> | JPEG + Blur           | 48.66 | 59.08 |
> | Gaussian Blur         | 48.17 | 57.78 |
> | JPEG + Noise          | 46.56 | 56.95 |
> | Noise                 | 43.44 | 52.60 |
> | JPEG                  | 42.56 | 51.02 |
> | None                  | 36.22 | 44.75 |
>
>
> As shown in the above table, the JPEG + Noise + Blur perturbation combination performs better on Magicbrush and SIDA than the previous best model, which used Noise and Blur. As it is a subset of the other model, we expect this model to be at least as good as our previous model. Interestingly, JPEG compression provides additional forensic information not captured by noise and blur, as evidenced by the approximately 2-point improvement in IoU and F1.
>
> The reason that we restricted ourselves to Gaussian Blur and Noise is because of the previous work of MINDER [1]. The authors argue that each perturbation has a blind spot, e.g. Gaussian blur works best on facial and noise on generic images due to noise being applied uniformly, thus the signal ends up mostly in the background.
>
> [1] Tsai, C. T., Ko, C. Y., Chung, I. H., Wang, Y. F., & Chen, P. Y. (2024). Understanding and Improving Training-Free AI-Generated Image Detections with Vision Foundation Models*. arXiv:2411.19117.
>
>
> >(W2): Failure case analysis: It would be helpful to include more examples of Detective SAM failure cases, particularly for fine-grained or challenging forgery categories.
>
> [AW2]: To resolve this, we create a qualitative figure on a failure case per dataset for Detective SAM: [Low IoU sample for each dataset with Detective SAM](https://pasteboard.co/ZNJZZp74S5Im.jpg). A common pitfall is Detective SAM selecting the wrong object in the vicinity of the forged one. A discussion with a reference to this figure will be added to the end of Section 4.1.
> In the current paper, we calculate the average IoU for all models per edit method (see Appendix F.4) to investigate whether failures are overrepresented per edit type. We observe that the 'Remove' edit is particularly difficult to localize.

---

> > ### Comment · Reviewer_miFc · 2025-11-26
> >
> > Thank the authors for the rebuttal responses and additional experiments. I think technically, the paper is of a relatively good quality for acceptance. I am keeping my positive rating for now, and will make final adjustments based on any ethical review feedback (if there will be any), as most of us flagged ethical concerns.

---

> > > ### Author Response · Authors · 2025-11-26
> > > **Thank you for the reply**
> > >
> > > Dear Reviewer miFc,
> > >
> > > Thank you for your positive response. If any concerns come up, please do not hesitate to ask further questions.
> > > We have added a general comment addressing the ethical concerns raised.
> > >
> > > Best regards,
> > >
> > > The Authors

---

### Official Review · Reviewer_EVaR · 2025-10-29

**Soundness:** 3
**Presentation:** 2
**Contribution:** 2
**Rating:** 4
**Confidence:** 3

**Summary:**

This paper proposes Detective SAM, a lightweight and adaptive framework for localizing forgeries in diffusion-edited images. Rather than relying on differences between clean and edited images, the method utilizes perturbation-induced embedding shifts extracted from a single edited image. These perturbation-based forensic signals are injected into a frozen SAM2 backbone using two lightweight modules: feature adapters that refine the decoder inputs, and a mask adapter that generates an automatic heatmap prompt to guide forgery segmentation. To maintain robustness against evolving editing techniques, the authors further introduce AutoEditForge, an automated pipeline that generates diverse and realistic diffusion-based forgeries for continual fine-tuning. Evaluations across seven datasets and seven baselines show that Detective SAM achieves strong out-of-distribution performance while enabling rapid adaptation to new editing models with minimal computational overhead.

**Strengths:**

1. The paper clearly identifies the limitations of existing forgery localization methods when applied to diffusion-edited images, and effectively addresses both the performance degradation and the challenge of adapting to continuously emerging editing models.
2. The proposed approach fully leverages the segmentation capabilities of SAM2 by introducing lightweight adapter modules that can be efficiently trained and reliably guide localization across a variety of recent diffusion-based editing pipelines.
3. Extensive experiments against seven competitive baselines across seven diverse datasets provide strong empirical evidence of the robustness and generalization performance of Detective SAM.

**Weaknesses:**

1. While the authors emphasize the importance of continual fine-tuning using AutoEditForge (Table 2) and set catastrophic forgetting as one of persistent problems in current IFL systems, it remains unclear whether the proposed adapter modules are genuinely resilient to catastrophic forgetting. The current evidence does not directly show whether older edit distributions remain preserved when new diffusion models are introduced.
2. The paper consistently refers to unseen datasets as “out-of-distribution” (OOD). However, the training set MagicBrush and the test set AutoSplice share the same editing model, DALL-E. Given that the diffusion models are trained to estimate training image distributions, the distinction between in-distribution and OOD becomes ambiguous.
3. The feature adapters are described as learning residual delta corrections between clean and perturbed embeddings, yet the paper provides limited analysis to verify what these corrections represent or how they contribute to performance improvements.
4. The manuscript repeatedly emphasizes the lightweight and efficient nature of Detective SAM. While this is valuable, the motivation for efficiency as a primary design goal is not entirely clear, given that forgery localization typically prioritizes accuracy over speed.
5. Although AutoEditForge is central to continual adaptation, the paper does not quantitatively evaluate the realism or consistency of its generated edits.

Note: Weaknesses 1-5 correspond directly to Questions 1-5.

**Questions:**

1. It would be helpful if the authors could clarify how the adapter design explicitly mitigates forgetting, especially considering that fine-tuning uses only 500 images from recent diffusion models. If future models generate substantially more new edits, would Detective SAM still maintain localization accuracy on prior model generations? A brief discussion or empirical analysis around Lines 81–83 could strengthen this point.
2. Could the authors clarify their OOD definition in this context? Specifically, when two different image inputs are processed by the same diffusion model (e.g., DALL-E), how is it justified to treat them as distinct distributions?
3. Including visualization, ablation, or interpretability results showing what the delta corrections capture--such as which regions or features are being adjusted--would make the role of this component clearer. A short qualitative example could effectively illustrate its necessity and benefit.
4. The authors could further clarify in what deployment scenarios the efficiency aspect is critical. For instance, is the method intended for frequent online updates or resource‑limited forensic systems? Discussing the practical motivation would help contextualize the design choice.
5. It would strengthen the paper if the authors could include either a quantitative assessment (e.g., perceptual quality metrics or human evaluation) or a qualitative justification for the realism and diversity of AutoEditForge outputs. This would better support the claim that the generated data effectively represents real‑world forgery scenarios.

---

> ### Author Response · Authors · 2025-11-18
> **Response to Reviewer EVaR (Q1, W1)**
>
> Thank you for your valuable comments and questions. Below, we respond to the weaknesses and questions highlighted.
>
> >(Q1, W1), Part 1:  It would be helpful if the authors could clarify how the adapter design explicitly mitigates forgetting, especially considering that fine-tuning uses only 500 images from recent diffusion models.
>
> [AQ1, AW1], Part 1: The adapter design mitigates forgetting by updating only the feature and mask adapters while the full SAM2 backbone remains frozen. Thus the task-agnostic SAM2 capabilities are preserved, while only tuning the task specific part. Fine-tuning with replay then allows the adapters to focus on the forgery task with new data. In Appendix F.3, we observe that there is catastrophic forgetting if no replay is used. Further, we note that fine-tuning on new data also improves OOD scores due to the data diversity. We will further clarify this in the paper in contribution 2 (line 95).
>
> >(Q1, W1), Part 2:  If future models generate substantially more new edits, would Detective SAM still maintain localization accuracy on prior model generations?
>
>  [AQ1, AW1], Part 2: To investigate your remark, we perform two new experiments: we fine-tune Detective SAM with 1.500 QWEN-Bench samples and 500 FLUX-Bench samples, and vice versa (with appropriate hyperparameter tuning). This allows us to analyze what happens in an unbalanced setting, where one diffusion model is overrepresented in the fine-tuning data.
>
> **Table E: Unbalanced Samples Fine-tuning Evaluation**
> | Test  Dataset | QWEN 1500, FLUX 500| FLUX 1500, QWEN 500 | FLUX 500, QWEN 500 |
> |-|-|-|-|
> | FLUX | 42.16/49.08 | **44.30/54.96** | 43.08/52.52 |
> | QWEN | **44.94/53.89** | 39.68/50.41 | 41.44/51.49 |
> | NanoBanana | 26.65/34.47 | 25.80/33.15 | **27.00/36.2**1 |
> | Magicbrush | 44.14/56.19 | **45.51**/56.83 | 45.03/**57.24** |
> | SIDA | 50.46/58.87 | 49.72/57.67 | **51.35/60.74** |
> | CoCoGLIDE | 44.95/55.40  | 42.57/53.12 | **45.37/55.62** |
> | AutoSplice | 43.30/54.97 | 44.31/**55.85** | **44.42**/57.02 |
>
> Notice that unbalanced fine-tuning improves performance for the overrepresented dataset but increases forgetting. This is possibly due to the fine-tuning samples' diversity being unbalanced with respect to the replay samples. Thus, if future models generate substantially more new edits, we would need to increase the replay or subsample the edits.
>
> The default setting for our periodic adaptation is 500 samples per set, with 20% replay. This is substantiated by the figure in Appendix F.3. Looking at the third column of the above table, this setting strikes a balance between forgetting and incremental adaptation.

---

> > ### Author Response · Authors · 2025-11-18
> > **Response to Reviewer EVaR (Q2, W2 - Q3, W3)**
> >
> > >(Q2, W2): Could the authors clarify their OOD definition in this context? Specifically, when two different image inputs are processed by the same diffusion model (e.g., DALL-E), how is it justified to treat them as distinct distributions?
> >
> > [AQ2, AW2]: Our definition of OOD is: "Completely unseen test sets for a fair comparison to baselines." (line 315).  Therefore, this relates not only to the model but also to dataset factors such as instruction types, image sources, edit types, mask sizes, and the editing model. To illustrate this, we can compare MagicBrush and AutoSplice:
> > 1. **Instruction:** Magicbrush contains action-oriented instructions from human crowd workers, whereas AutoSplice has descriptive captions, generated by modifying the image caption.
> > 2. **Data source:** Magicbrush uses images from MS COCO [1] and AutoSplice uses Visual News [2].
> > 3. **Edit types:** Magicbrush contains semantic changes and AutoSplice contains mainly insertions and replacements.
> > 4. **Edit sizes:** 84% of Magicbrush edit masks cover less than 25% of the image, whereas 68% of AutoSplice masks occupy more than 25% (Appendix G.1).
> > 5. **Editing model:** Both Magicbrush and AutoSplice use DALL-E 2.
> >
> > We believe that AutoSplice's high scores are mainly due to its large mask sizes, since TruFor, PSCC-Net and Mesorch all achieve their highest scores on this dataset. We will clarify our OOD definition in the main text by listing these properties.
> >
> > To recover editor diversity, we add a new OOD benchmark: UltraEdit (below).
> > Further, we add a table to the appendix with all datasets and their editing models:
> >
> > | Dataset       | Editing Model / Method  |
> > |-|-|
> > | MagicBrush    | DALL-E 2                |
> > | SIDA          | Latent Diffusion Model  |
> > | AutoSplice    | DALL-E 2                |
> > | CoCoGLIDE     | GLIDE                   |
> > | NanoBanana    | Gemini 2.5 Flash        |
> > | FLUX-Bench    | FLUX Kontext            |
> > | QWEN-Bench    | QWEN-Image-Edit         |
> > | UltraEdit     | StableDiffusionXL-Turbo |
> >
> > We perform an additional experiment on the recent UltraEdit [3] dataset (SDXL-Turbo) with all baselines, using the region-based subset of the dataset, consisting of 100.000 samples.
> > Below are the IoU/F1 scores for all models on UltraEdit.
> >
> > **Table F: UltraEdit Baseline Comparison**
> > | Model          | UltraEdit IoU / F1    |
> > |-|-|
> > | SAFIRE         | 18.41 / 24.00 |
> > | Mesorch        | 5.45 / 7.51   |
> > | AdaIFL         | 7.73 / 11.23  |
> > | TruFor         | 16.15 / 22.32 |
> > | SIDA           | 3.29 / 4.45   |
> > | FakeShield     | 12.98 / 18.32 |
> > | PSCC-Net       | 10.06 / 15.43 |
> > | DetectiveSAM   |**27.74 / 35.54** |
> >
> > Analyzing the above table, we can see that Detective SAM also generalizes to UltraEdit and that SIDA and TruFor perform relatively well.
> > We also provide qualitative results of the predictions on UltraEdit with [mask predictions for all models on 4 UltraEdit samples.](https://pasteboard.co/NlVD7bheUmyx.png)
> > We will extend Section 4.1 with this explanation and add a note to Table 1 regarding the similarity of the editing model in the final PDF.
> >
> >
> > [1] Lin, T. Y., Maire, M., Belongie, S., Bourdev, L., Girshick, R., Hays, J., Perona, P., Ramanan, D., Zitnick, C. L., and Dollár, P. (2015). Microsoft COCO: Common Objects in Context. arXiv:1405.0312.
> >
> > [2] Liu, F., Wang, Y., Wang, T., and Ordonez, V. (2021). Visual News: Benchmark and Challenges in News Image Captioning. arXiv:2010.03743.
> >
> > [3] Zhao, H., Ma, X., Chen, L., Si, S., Wu, R., An, K., Yu, P., Zhang, M., Li, Q., and Chang, B. (2024). UltraEdit: Instruction based Fine Grained Image Editing at Scale. arXiv:2407.05282.
> >
> >
> > >(Q3, W3): A short qualitative example showing what the delta corrections capture could effectively illustrate its necessity and benefit.
> >
> > [AQ3, AW3]: We agree that this is an interesting approach to analyze and therefore compute a saliency map visualization of the delta corrections to analyze the captured features. The saliency maps of the Delta corrections, averaged over the embedding dimension and bilinearly upsampled to 512 X 512, are provided with [4 Magicbrush samples](https://pasteboard.co/iRu5wLaJjqvP.png) and [4 CoCoGLIDE samples](https://pasteboard.co/WVlw9IhyWBRx.png).
> >  We notice that the Delta corrections capture contours of existing objects and the magnitude is generally either high or low for the final predicted masks. It is hard to discern distinct properties from these maps, but they show that complex patterns are captured in each feature map resolution.

---

> ### Author Response · Authors · 2025-11-18
> **Response to Reviewer EVaR (Q4, W4 - Q5, W5)**
>
> >(Q4, W4): The authors could further clarify in what deployment scenarios the efficiency aspect is critical.
>
> [AQ4, AW4]: We provide an overview of deployment scenarios and a table showing the throughput of CoCoGLIDE, to demonstrate the practical difference in efficiency between models.
> 1. Workflows where many images must be screened at scale or in resource-limited environments such as on platforms. Serving Detective SAM cheaply allows for widescale spread of the benefits among consumers via online tools.
> 2. We have observed that existing IFL systems collapse under state-of-the-art editors. This automatically raises the need for IFL systems to stay up to date. Periodic adaptation bridges the gap between the academic reality, where beating the editors until the paper's release is important, and the practical reality. IFL systems need frequent updates to be useful; therefore, this process must be efficient.
>
> Further, we provide the quantitative evidence of the first statement. We demonstrate the difference in inference efficiency using a throughput experiment on 512 CoCoGLIDE samples, with a batch size of 1 and no parallel processing, on an NVIDIA H100:
>
> **Table B: Throughput and Inference Time Baseline Comparison**
> | Metric|SAFIRE| Mesorch | AdaIFL | TruFor | SIDA   | FakeShield | PSCC-Net | DetectiveSAM |
> | -| - | -| - | - | - | -  | - | - |
> | Images per second        | 0.35    | 37.10   | 9.54   | 22.61  | 2.50   | 2.67       | 77.70    | 29.04        |
> | Total inference time (s) | 1475.04 | 13.80   | 53.66  | 22.64  | 204.50 | 191.54     | 6.59     | 17.63        |
> | Average OOD IoU | 24.11 | 24.73 | 27.90 | 17.43 | 17.55 | 15.68 | 26.99 | 36.99 |
>
>
> As shown in the table above, the MLLM methods take several minutes, whereas SAFIRE takes around half an hour with this limited number of samples. This makes practical usage difficult.
>
> >(Q5, W5): It would strengthen the paper if the authors could include either a quantitative assessment (e.g., perceptual quality metrics or human evaluation) or a qualitative justification for the realism and diversity of AutoEditForge outputs.
>
> [AQ5, AW5] We introduce a quantitative evaluation with the perceptual quality metrics BRISQUE [1], NIQE [2], and PI [3] and add a qualitative figure of each AutoEditForge dataset. The metrics are non-reference, since reference metrics measure similarity, which is directly biased by the mask size. Therefore, we compute the metrics for the source and tampered datasets to investigate the difference. For these metrics, lower is better.
> [BRISQUE, NIQE, PI perceptual quality metrics for all datasets with (source, tampered) pairs available.](https://pasteboard.co/lQSGaSjGyCir.png) As can be seen in the figure, the differences between the source and tampered images are small for the AutoEditForge datasets (FLUX, QWEN and NanoBanana) and CoCoGLIDE, but noticeable for Magicbrush and AutoSplice. This confirms that AutoEditForge shows no significant degradation in quality with respect to the source images.
>
> Next, we create a figure with the first three (source, tampered) pairs of the AutoEditForge datasets: [AutoEditForge qualitative figure](https://pasteboard.co/tXpruvXyrvOi.png). We can see altered objects (top left), colour changes (middle), objects that have been removed (top right) and objects that have been added (bottom left).
>
> The PDF will be updated to include the perceptual quality investigation and the relevant discussion.
>
> [1] A. Mittal, A. K. Moorthy, and A. C. Bovik, "No-reference image quality assessment in the spatial domain," IEEE Transactions on Image Processing, vol. 21, no. 12, pp. 4695-4708, 2012.
> [2] A. Mittal, R. Soundararajan, and A. C. Bovik, "Making a 'completely blind' image quality analyzer," IEEE Signal Processing Letters, vol. 20, no. 3, pp. 209-212, 2013.
> [3] Blau, Y., Mechrez, R., Timofte, R., Michaeli, T., and Zelnik-Manor, L. The 2018 PIRM Challenge on Perceptual Image Super-resolution. Proceedings of the European Conference on Computer Vision Workshops, 2018.

---

### Official Review · Reviewer_De2b · 2025-10-31

**Soundness:** 3
**Presentation:** 3
**Contribution:** 2
**Rating:** 6
**Confidence:** 5

**Summary:**

This paper presents Detective SAM, a framework that localizes AI forgeries by detecting feature shifts in a perturbed image using a frozen SAM2 model. It is paired with AutoEditForge, an automated pipeline that generates up-to-date training data. This dual approach allows the model to significantly outperform existing methods and rapidly adapt to counter new, state-of-the-art image editors.

**Strengths:**

The paper's primary strength is its methodological design, which leverages the powerful SAM2 foundation model for the task of forgery localization. Rather than designing a new architecture, the authors integrate a perturbation-driven forensic clue using lightweight feature adapters and an automatic mask adapter.  This design is effectively paired with the AutoEditForge pipeline for automated data generation. The adaptable model and data pipeline present a structured approach to the challenge of evolving forgery techniques.

**Weaknesses:**

1. While this work performs very well, it appears more like a system-level optimization from an engineering perspective, particularly the AUTOEDITFORGE component. I do not object to leveraging the capabilities of existing models to improve a specific task, but the authors' overall framework seems more like an integration of existing work. For instance, numerous studies have explored perturbations at both the image and feature levels, and the concept of feature adapters is also well-established. I would like the authors to discuss the scientific contribution of this paper.

2. The method is tightly coupled with the promptable architecture of SAM2. While this is a clever design choice, it raises critical concerns about the generality of the proposed paradigm. It is unclear whether this 'perturb-adapt-prompt' approach can be successfully transferred to other prompt-based segmentation models, or if its success is inextricably linked to the specific properties of SAM2's decoder. The paper would be strengthened by a discussion on the conditions required for this transferability, or an explanation of why it might be limited to this specific model family.

3. Using image perturbations to distinguish manipulated regions is a proven and effective method. However, when external noise is added to an edited image, it is unclear whether this approach can maintain its consistent generalization performance.

**Questions:**

1. The discussion around Tab 3(a) feels insufficient. Could the authors provide more intuition as to why the combination of Gaussian Noise and Blur emerges as the optimal pairing? Given the paper's claim that  the perturbation type has a significant impact on
the localization performance, it would be beneficial to see experiments with a more diverse set of perturbation types to fully substantiate this argument.

2. How would the proposed method perform in a classic forgery scenario where an image is first subtly edited (not visible to the naked eye) and then globally post-processed with transformations like Gaussian noise to mask the tampering artifacts? Is the model robust enough to distinguish its own diagnostic perturbations from pre-existing, confounding noise in the image?

3. Could you clarify the evaluation protocol regarding the results in Table 1? The fine-tuned Detective SAM_{SOTA} is presented in Table 2 as the final version of the model, yet it is not benchmarked against the initial baselines in Table 1.

4. What quality control mechanisms are in place to ensure the usability of images generated by AUTOEDITFORGE? Could you provide statistics on the pipeline's reliability, such as the approximate proportion of automatically generated samples that require manual intervention or are discarded? Furthermore, what are the common failure modes of this generation process?

---

> ### Author Response · Authors · 2025-11-18
> **Response to Reviewer De2b (W1-W3)**
>
> Thank you for your valuable comments and questions. Below, we respond to the weaknesses and questions highlighted.
>
> >(W1): The work appears more like a system-level optimization. I would like the authors to discuss the scientific contribution of this paper.
>
> [WA1]: We summarize the key scientific contributions in three points:
> 1. **We conjecture and verify the performance collapse of IFL systems:**  Methods that were state-of-the-art at the time of their release, collapse under the newest edits. Verifying this statement required the creation of state-of-the-art datasets, resulting in the development of AutoEditForge. Furthermore, we present Detective SAM as a practical method to the problematic consequences of this insight.
> 2. **We explore explicit perturbations in a learned system:** Explicit perturbations have been used in a training-free manner and implicit perturbation signals have been successful in systems such as CAT-NET [1] for legacy forgery types. Using explicit perturbation signals in a learned system with adapters is a novel approach.
> 3. **We fabricate the architecture to use SAM 2 efficiently for IFL:** The mask adapter differs from the only other learned SAM prompter (IMDPrompter, Section 2). Combining these components is not trivial, but crucial to creating an effective system.
>
> Thus, we believe that the scientific contribution stems from the model collapse insight, solved by the novel use of an explicit perturbation signal and a uniquely efficient automatic prompter, as well as AutoEditForge, which supports adaptation to new editors and further advances the field.
>
> [1] Myung-Joon Kwon, In-Jae Yu, Seung-Hun Nam, and Heung-Kyu Lee. CAT-Net: Compression Artifact Tracing Network for Detection and Localization of Image Splicing, Proceedings of the IEEE/CVF Winter Conference on Applications of Computer Vision (WACV)
>
> >(W2): The paper would be strengthened by a discussion on the conditions required for this transferability of the "perturb-adapt-prompt" approach, or an explanation of why it might be limited to this specific model family.
>
> [WA2]: The method leverages SAM2 because its promptable decoder, downstream adaptation results [1] and strong encoder make it suitable for the use of explicit perturbation clues. Besides this, we also utilize SAM2 because it is the state-of-the-art segmentation model at the time of writing this paper. The components that truly depend on SAM2 are narrow since prior work [2] has shown similar perturbation sensitivity in other encoders.
>
> The architectural requirements are:
>  1. A state-of-the-art vision encoder.
>  2. A promptable and adaptable segmentation decoder.
>  3. The decoder being trained jointly with the encoder.
>
> Without this encoder–decoder alignment, the Delta corrections also need to bridge the representational mismatch, which would undermine the benefit of lightweight adapters.
> Thus we limit the 'perturb-adapt-prompt' approach to models with a matched state-of-the-art encoder and segmentation decoder. Thank you for raising this interesting point.
>
> [1] Chen, T., Lu, A., Zhu, L., Ding, C., Yu, C., Ji, D., Li, Z., Sun, L., Mao, P., & Zang, Y.
> SAM2-Adapter: Evaluating & Adapting Segment Anything 2 in Downstream Tasks: Camouflage, Shadow, Medical Image Segmentation, and More. arXiv:2408.04579, 2024.
>
> [2] Tsai, C. T., Ko, C. Y., Chung, I. H., Wang, Y. F., & Chen, P. Y. (2024). Understanding and Improving Training-Free AI-Generated Image Detections with Vision Foundation Models. arXiv:2411.19117.
>
>
> >(W3, Q2):  When external noise is added to an edited image, it is unclear whether this approach can maintain its performance.
>
> [AW3, AQ2]: To analyze this, we perform an additional experiment and apply Gaussian Blur, Gaussian Noise and JPEG Compression to the images of CoCoGLIDE for three increasing intensities. We restrict ourselves to the five best performing models. This yields a graph of the [relative IoU change](https://pasteboard.co/H9VXZrra7XgT.png) and the [absolute IoU](https://pasteboard.co/EpXiHFcw27fL.png) over increasing perturbation intensities.
>
> Analyzing the figures, we note that Detective SAM is relatively robust to Gaussian Blur and Noise. These are the perturbations used as forensic clues. We speculate that the model is trained to interpret the difference in embedding shifts. This means that external noise added to the input affects both streams in a similar way, and the delta between them remains informative. Detective SAM is in line with the baselines in terms of robustness to JPEG compression.

---

> ### Author Response · Authors · 2025-11-18
> **Response to Reviewer De2b (Q1-Q4)**
>
> >(Q1): Could the authors provide more intuition as to why the combination of Gaussian Noise and Blur emerges as the optimal pairing?
> >
> [AQ1]: To directly answer your question, we investigate every combination of perturbations and extend Table 3(a) in the paper as an additional experiment:
>
> **Table C: Extending Detective SAM's Forensic Perturbations**
> | Perturbation          | IoU ↑ | F1 ↑  |
> |-|-|-|
> | JPEG + Noise + Blur   | 52.58 | 63.08 |
> | Noise + Blur          | 50.52 | 61.42 |
> | JPEG + Blur           | 48.66 | 59.08 |
> | Gaussian Blur         | 48.17 | 57.78 |
> | JPEG + Noise          | 46.56 | 56.95 |
> | Noise                 | 43.44 | 52.60 |
> | JPEG                  | 42.56 | 51.02 |
> | None                  | 36.22 | 44.75 |
>
> As shown in the above table, the JPEG + Noise + Blur perturbation combination performs better on Magicbrush and SIDA than the previous best model, which used Noise and Blur. As it is a subset of the other model, we expect this model to be at least as good as our previous model. Interestingly, JPEG compression provides additional forensic information not captured by Noise and Blur, as evidenced by the approximately 2-point improvement in IoU and F1.
>
> To provide intuition on our initial choice, we restricted ourselves to Noise and Blur because of the previous work of MINDER [1]. The authors argue that each perturbation has a blind spot, e.g. Gaussian blur works best on facial and noise on generic images due to noise being applied uniformly, thus the signal ends up mostly in the background.
>
> [1] Tsai, C. T., Ko, C. Y., Chung, I. H., Wang, Y. F., & Chen, P. Y. (2024). Understanding and Improving Training-Free AI-Generated Image Detections with Vision Foundation Models. arXiv:2411.19117.
>
> >(Q3): Could you clarify the evaluation protocol regarding the results in Table 1? The fine-tuned Detective SAM_{SOTA} is presented in Table 2 as the final version of the model, yet it is not benchmarked against the initial baselines in Table 1.
>
> A: Detective SAM$^{\text{SOTA}}$ is benchmarked against the baselines in Table 1; in Appendix F.2, which is referenced at line 431. We notice an increased performance over these baselines compared to Detective SAM without finetuning and, mostly driven by increased performance in NanoBanana, attribute this to the exposure to modern FLUX and QWEN data.
> For reference, we copy Table 1 and the appendix table below:
>
> | Model | MagicBrush | SIDA | CoCoGLIDE | AutoSplice | NanoBanana | Avg OOD |
> | - | - | - | - | - | - | - |
> | Detective SAM | 46.48/57.55 | 54.55/65.29 | 44.74/51.50 | 46.90/60.30 | 19.34/20.77 | 36.99/44.19 |
> | Detective SAM$^{\text{SOTA}}$ | 45.03/57.24 | 51.35/60.74 | 45.37/55.62 | 44.42/57.02 | 27.00/36.21 | 38.93/49.62 |
>
> >(Q4): What quality control mechanisms are in place to ensure the usability of images generated by AUTOEDITFORGE? Could you provide statistics on the pipeline's reliability and common failure modes?
>
> [AQ4] AutoEditForge implements several quality control mechanisms that were tracked during the generation process, but were only lightly documented in the paper.
>
> - Multi-metric duplicate detection: Four complementary metrics (blob analysis, MAE, pHash and SSIM) are used to validate meaningful inpainting changes and automatically reject failed images without retrying.
> - Mask validation pipeline: All masks undergo format validation, size matching, and area constraint checks, ensuring only high-quality masks proceed to inpainting.
> - An error tracker categorizes failures across 11 error types.
>
> We will discuss this in more detail in the appendix. Further, we compose a table analyzing the error logs of FLUX-Bench and QWEN-Bench, totalling 6.000 samples.
> In total 9.446 images were generated, with 3.443 failures, giving a failure rate of 36.45%. Each editing method has 25% of the images due to our class balancing (Appendix D).
> The failures are distributed as follows:
>
> **Table D: AutoEditForge Failure Categories**
> | Failure mode                           | Count | % of errors |
> |-|-:|-:|
> | Inpainting produced no result          | 1730  | 50%   |
> | Florence mask coverage validity        | 1506  | 44%   |
> | Florence captioning failed             | 187   | 5%    |
> | LLM object selection failure           | 17    | 0.5%  |
> | SAM segmentation mask failure          | 1     | 0.03% |
> | Fallback mask file creation errors     | 1     | 0.03% |
> | LLM edit method decision failures      | 1     | 0.03% |
>
> Error tracking logs reveal two main types of failure:
> 1. Inpainting model fails to modify the image: The model produces outputs nearly identical to inputs despite valid masks/prompts. Our multi-metric duplicate detection automatically catches and rejects these cases.
> 2. Inappropriate object selection: The LLM occasionally selects objects that are too small or too large for meaningful edits. Our mask validation pipeline rejects these before inpainting.

---

### Official Review · Reviewer_sKpy · 2025-11-03

**Soundness:** 3
**Presentation:** 3
**Contribution:** 3
**Rating:** 6
**Confidence:** 3

**Summary:**

The paper proposes Detective SAM, a framework for image forgery localization that adapts the Segment Anything Model 2 (SAM2) to detect and precisely mark regions of AI-generated edits. Instead of retraining SAM2, the authors freeze its backbone and attach small trainable “feature” and “mask” adapters that use differences between an image and its perturbed versions (blurred, noised, or compressed) as forensic clues indicating manipulation. These perturbation responses are converted into an automatic heatmap prompt guiding SAM2 to segment the tampered areas. To keep pace with rapidly changing diffusion-based editors, they also introduce AutoEditForge, an automated pipeline that continually produces realistic, labeled image edits (Replace, Remove, Add, Change Partially) for fine-tuning and evaluation. Together, these components aim to build an adaptive, lightweight, and continually updatable system that localizes forgeries across diverse generative-model edits more reliably than prior detectors.

**Strengths:**

1- The integration of perturbation-driven forensic signals with SAM2 is technically neat and explained with adequate mathematical detail.

2- Results across seven datasets demonstrate stable performance under moderate domain shift, which is rare among IFL systems.

**Weaknesses:**

1- In Abstract & Sec. 3.3, the system is framed as a continual, lifelong learning pipeline that updates smoothly as new editors appear. But in Sec. 4.1–4.2, the experiments are all offline fine-tuning on small fixed batches (500 samples), without any demonstration of online or streaming adaptation. The “continual” claim is thus aspirational rather than empirically validated.

2- In Abstract & Intro, Detective SAM supposedly handles arbitrary new diffusion editors “without retraining.” but in Sec. 4 in the Training specification, the model is trained and tested only on diffusion-edited datasets (SIDA, MagicBrush, AutoSplice, CoCoGLIDE, NanoBanana) and fails sharply on unseen SOTA editors until retrained. The paper’s own results contradict the “no-retraining” implication.

3-  In Sec. 3.2, they emphasize lightweight adapters—81 k + 887 k parameters—trained in “two hours on an H100.” but in Sec. 4 Baselines, inference comparisons are made against massive MLLMs but all are run on a single H100 GPU, ignoring throughput, preprocessing (multiple perturbations per image), and repeated SAM2 encodings. The real computational cost per sample is much higher than implied; “lightweight” refers only to trainable weights, not runtime.

**Questions:**

Your method is described as a general framework for image forgery localization, yet the core signal it relies on is perturbation-induced embedding instability (Section 3.1–3.2). This assumes that forged regions respond differently to Gaussian noise, blur, or compression compared to authentic ones. However, not all forgeries—e.g., copy-move, GAN inpainting, or carefully composited human edits—necessarily exhibit such perturbation sensitivity. Could you clarify whether Detective SAM is actually detecting forgery semantics or merely perturbation sensitivity, and how the method would generalize to forgery types that do not produce perturbation-based cues?

This question matters because, the authors blur the distinction between “detecting forgeries” (a semantic problem) and “detecting perturbation-sensitive regions” (a statistical proxy). The two overlap for diffusion artifacts but diverge for other forgery classes. Hence, the approach is domain-specific rather than general, despite the paper’s repeated framing as a “comprehensive framework for modern IFL.”

**Details Of Ethics Concerns:**

No major concerns on ethics, but given the topic and application (image forgery), maybe worth a look.

---

> ### Author Response · Authors · 2025-11-18
> **Response to Reviewer sKpy (W1 - W2)**
>
> Thank you for your valuable comments and questions. Below, we respond to the weaknesses and questions highlighted.
>
> >(W1): The system is framed as a continual, lifelong learning pipeline that updates smoothly as new editors appear. But the experiments are all offline fine-tuning on small fixed batches (500 samples), without any demonstration of online or streaming adaptation.
>
>
> [AW1]: Our “continual adaptation” framework arises from the observed severe performance degradation of existing IFL systems on modern image editors. Focusing on practical applications, a system should stay continually up to date. We agree that this is unlike online learning or streaming adaptation. Detective SAM can ingest new editors with little effort, using AutoEditForge to generate new samples and fine-tune the system via replay. Thus, we also note that continual is used here in a different sense and we should have used a different term, say, “Periodic adaptation”. The paper investigates this property using Table 2, in which we fine-tune state-of-the-art editors using a limited number of samples in order to improve performance.
>
> To support our claim, we conduct an additional experiment. We fine-tune Detective SAM incrementally on 500 FLUX-Bench samples and then on 500 QWEN-Bench samples, and vice versa.
>
> **Table A: Evaluation of Incremental Fine-tuning**
> | Dataset | Detective SAM | FLUX 500 | QWEN 500 | FLUX 500 -> QWEN 500 | QWEN 500 -> FLUX 500|
> |-|-|-|-| - | - |
> | FLUX | 18.70/21.28 | 41.09/52.81 | 29.60/39.66 |  41.43/51.25 | 43.34/53.68
> | QWEN | 20.41/22.29 | 32.26/43.05 | 42.43/53.93 | 43.20/52.72  | 42.58/54.19
> | Average Table 1 OOD* | 36.99/44.19 | 35.90/45.63 |  34.47/44.09  | 37.68/48.24 | 36.95/47.96
>
> *includes CoCoGLIDE, AutoSplice and NanoBanana
>
> Examining the above table, we can see that for sequential fine-tuning, the first 500 new editor samples slightly reduce OOD performance, which is then restored when the next dataset is used for tuning. Overall, the similarity between the sequential fine-tuning results and those of Detective SAM$^{\text{SOTA}}$ in Table 2 suggest that such adaptation is effective over more than one step.
> We will clarify the usage of the term "continual" in the introduction.
>
> >(W2): In Abstract & Intro, Detective SAM supposedly handles arbitrary new diffusion editors “without retraining.” but in Sec. 4 in the Training specification, the model is trained and tested only on diffusion-edited datasets (SIDA, MagicBrush, AutoSplice, CoCoGLIDE, NanoBanana) and fails sharply on unseen SOTA editors until retrained. The paper’s own results contradict the “no-retraining” implication.
>
> [AW2]: Thank you for highlighting this. We do not intend to claim "without retraining", but efficient adaptation. The reference to “training-free” in the introduction concerns the perturbation-based forensic signal, which does not require learned filters (line 49). In the abstract, we emphasise Detective SAM's efficient adaptation. In the introduction, we state that "rapid progress in generative models creates a moving target that requires up-to-date data and training", and we relate this to the fine-tuning of Detective SAM in contribution 2 (line 94).
>
> Detective SAM is therefore not positioned as a static method without retraining. Instead, the key contribution is that the backbone remains frozen and only lightweight adapters are updated, making adaptation fast and practical when new editors emerge. We agree that this distinction can be stated more clearly, and we will revise the abstract wording to explicitly indicate that “updates” refer to efficient fine-tuning rather than a static process.

---

> > ### Author Response · Authors · 2025-11-18
> > **Response to Reviewer sKpy (W3 - Q1)**
> >
> > >(W3): In Sec. 3.2, they emphasize lightweight adapters, but in Sec. 4 Baselines, inference comparisons are made against massive MLLMs but all are run on a single H100 GPU, ignoring throughput, preprocessing (multiple perturbations per image), and repeated SAM2 encodings. The real computational cost per sample is much higher than implied; “lightweight” refers only to trainable weights, not runtime.
> >
> > [AW3] Detective SAM is lightweight in terms of trainable weights, which is essential for adaptation. The "lightweight" does not only apply to trainable weights:
> >
> > 1. To analyze the throughput, we create a new experiment, examining throughput within the respective forward passes for all 512 CoCoGLIDE samples without preprocessing:
> >
> > **Table B: Throughput and Inference Time Baseline Comparison**
> > | Metric|SAFIRE| Mesorch | AdaIFL | TruFor | SIDA   | FakeShield | PSCC-Net | Detective SAM |
> > | -| - | -| - | - | - | -  | - | - |
> > | Images per second        | 0.35    | 37.10  | 9.54   | 22.61  | 2.50   | 2.67       | 77.70    | 29.04   |
> > | Total inference time (s) | 1475.04 | 13.80 | 53.66  | 22.64  | 204.50 | 191.54     | 6.59     | 17.63   |
> > | Average OOD IoU | 24.11 | 24.73 | 27.90 | 17.43 | 17.55 | 15.68 | 26.99 | 36.99 |
> >
> > As shown in the above table, the MLLM methods take several minutes, whereas SAFIRE takes around half an hour with this limited number of samples. This makes practical usage difficult. We note that Detective SAM has the third fastest throughput, processing the dataset in 17.63 seconds, substantiating the "lightweight" claim.
> >
> > 2. Next, we profile the mean non-parallel preprocessing time per sample on the CPU for the 512 CoCoGLIDE samples. This gives us a mean preprocessing time of **17.20 ms** with a standard deviation of **3.98**. As the perturbations are straightforward and the processing can be parallelized, the preprocessing adds negligible overhead.
> >
> >
> > >(Q1): Your method is described as a general framework for image forgery localization, yet the core signal it relies on is perturbation-induced embedding instability. This assumes that forged regions respond differently to Gaussian noise, blur, or compression compared to authentic ones. However, not all forgeries necessarily exhibit such perturbation sensitivity. Could you clarify whether Detective SAM is actually detecting forgery semantics or perturbation sensitivity, and how it would generalize to forgery types that do not produce perturbation-based cues?
> >
> > >This question matters because, the authors blur the distinction between “detecting forgeries” (a semantic problem) and “detecting perturbation-sensitive regions” (a statistical proxy).  Hence, the approach is domain-specific, despite the paper’s repeated framing as a “comprehensive framework for modern IFL.”
> >
> > [AQ1] Thank you for raising this valid concern. Indeed, Detective SAM is not focused on semantics. We will change the wording in the introduction so that it no longer states 'comprehensive', as we only focus on artifacts, and will instead use the word 'practical'. This better reflects Detective SAM's objective.
> > MLLM methods are better suited for semantics. However, we believe that meaningful, harmful modern edits need to be semantically consistent since this is essential for adversaries to convince users. In this semantically consistent modern IFL landscape, we also have to consider artifacts, so "practical" is a more appropriate term.
> > It is therefore not evident that forgery detection is purely a semantic task, since the aim of forgery is typically to remain semantically consistent and legacy IFL edits such as copy-move were previously detected with statistical signals (CAT-NET [1], TruFor [2]).
> >
> > Regarding forgery types, copy-move and human edits (e.g. Photoshop) are harmful, but not within the focus of Detective SAM. Detective SAM focuses on diffusion edits that allow for automated, realistic and harmful edits. As the capabilities and ease of use of diffusion editors in tools such as Photoshop increase, human edits also utilize such models.
> > Therefore, we believe that Detective SAM is complementary to semantic-based forensic tools.
> >
> > A new study is needed to determine whether perturbation sensitivities persist in new paradigms such as GANs or specific autoregressive image editing architectures. There is empirical evidence in MINDER [3] that Gaussian Blur sensitivities also exist in GAN-generated images.
> >
> > [1] Myung-Joon Kwon, In-Jae Yu, Seung-Hun Nam, and Heung-Kyu Lee. CAT-Net, Proceedings of the IEEE/CVF Winter Conference on Applications of Computer Vision (WACV)
> >
> > [2] Guillaro, F., Cozzolino, D., Sud, A., Dufour, N., & Verdoliva, L. TruFor: Leveraging all-round clues for trustworthy image forgery detection and localization (2023). arXiv:2212.10957
> >
> > [3] Tsai, C. T., Ko, C. Y., Chung, I. H., Wang, Y. F., & Chen, P. Y. (2024). Understanding and Improving Training-Free AI-Generated Image Detections with Vision Foundation Models. arXiv:2411.19117.

---

### Author Response · Authors · 2025-11-18
**Response Summary by Authors**

We are grateful to all reviewers for their feedback and questions. We conducted several additional experiments with new figures to strengthen our paper. All updates will be added to the revised PDF.

The experiments are listed in the table below. Since there were several overlapping questions, some experiments are intended for multiple reviewers.

| Experiment | Reviewer(s) |
|-|-|
| Incremental fine-tuning (Table A)| sKpy W1 |
| Throughput and preprocessing profiling (Table B)| sKpy W3, EVaR Q4 & W4|
| Perturbation combinations extension (Table C) |De2b Q1, miFc W1 |
| Robustness line-plot ([relative](https://pasteboard.co/H9VXZrra7XgT.png), [absolute](https://pasteboard.co/EpXiHFcw27fL.png))| De2b Q2 & W3 |
| AutoEditForge reliability and failure modes (Table D)| De2b Q4 |
| Unbalanced fine-tuning (Table E) | EVaR Q1 & W1 |
| OOD clarification new dataset (Table F, [qualitative](https://pasteboard.co/NlVD7bheUmyx.png)) | EVaR Q2 & W2 |
| Delta correction saliency maps ([Magicbrush](https://pasteboard.co/iRu5wLaJjqvP.png), [CoCoGLIDE](https://pasteboard.co/WVlw9IhyWBRx.png)) | EVaR Q3 & W3 |
| AutoEditForge [perceptual quality metrics](https://pasteboard.co/lQSGaSjGyCir.png) and [qualitative figure](https://pasteboard.co/tXpruvXyrvOi.png) | EVaR Q5 & W5 |
| Qualitative Detective SAM [failure cases](https://pasteboard.co/ZNJZZp74S5Im.jpg) | miFc W2 |


We hope the experiments and clarifications effectively address the reviewers' concerns.

---

> ### Author Response · Authors · 2025-11-25
> **Revised draft**
>
> We have now updated the draft and rebuttal text to include the new experimental results, figures and additional descriptions.
>
> As reviewers raise concerns for an ethics review, we briefly discuss those concerns:
> 1. **Legal compliance:** AutoEditForge uses images from Open-Images V7 (CC BY 2.0 License), and all benchmark datasets are publicly available. The editing models for AutoEditForge are: Gemini 2.5 Flash, Flux Kontext, and Qwen-Image-Edit, which all permit and encourage academic use.
> 2. **Privacy, security and safety:** We mitigate risks by training solely on publicly available and synthetic datasets to ensure no new personal data is collected, and with this paper target an increase in a secure and safe online environment through the detection of forgery.
> 3. **Responsible research practice:** We promote reproducibility in our research by releasing our code and weights, in line with responsible research practice. The datasets will be released upon acceptance. Our automated dataset pipeline relies on publicly available data, eliminating the need for new human subjects or for annotators to provide uncompensated labour.
>
> We kindly ask the reviewers whether we have addressed their concerns sufficiently and welcome further discussion.

---

### Author Response · Authors · 2025-11-28
**Comment from the Authors to the new AC**

Dear (newly) Assigned AC,

In light of the decision to revert to the pre-discussion state, we provide a summary of our interactions and the experiments added to address the reviewer's concerns.

1. **Reviewer sKpy (6, Conf: 3):** Concerns regarding continual learning and efficiency of Detective SAM were addressed via new experiments in Table A (Incremental fine-tuning) and Table B (Throughput), as well as by profiling the pre-processing time and amending the continual learning text.
2. **Reviewer De2b (6, Conf: 5):** Questions regarding perturbation choices and dataset quality were resolved via new Table C (Perturbation Combinations) and Table D (AutoEditForge Failure Analysis), as well as conducting a robustness experiment over increasing Gaussian Blur, Noise, and JPEG compression intensities.
3. **Reviewer EVaR (4, Conf: 3):** The primary concern regarding our Out-Of-Distribution (OOD) definition was addressed by benchmarking on an entirely new dataset (UltraEdit) and clarifying the differences between OOD datasets.  New experiments were performed on unbalanced fine-tuning and perceptual quality. Further, at the reviewers' request, we generated feature visualizations for eight samples.
4. **Reviewer miFc (6, Conf: 4):** We provided the requested experiments on Detective SAM failure cases and perturbation choices. The reviewer replied: "I think technically, the paper is of a relatively good quality for acceptance," and maintained the positive rating.

The rebuttal contributions are all included in the revised PDF.
We have received no follow-up correspondence from Reviewers sKpy, De2b, and EVaR prior to the suspension.

Best regards,

Authors of Detective SAM

---

### Meta-Review · Area_Chair_cJrY · 2025-12-21

**Summary:**

This paper presents Detective SAM, a framework based on SAM2 for image forgery localization to detect regions edited by diffusion-based editors. Detective SAM utilizes perturbation-driven feature embeddings as forensic signals and incorporates feature adapters to refine decoder inputs along with a mask adapter that automatically generates heatmap prompts for the frozen SAM2 backbone. The authors also introduce AutoEditForge to generate up-to-date training data, enabling continual adaptation of Detective SAM through efficient fine-tuning. Experiments demonstrate that Detective SAM achieves competitive performance across multiple benchmarks while maintaining efficiency in adapting to new editing techniques.

The paper received initial scores of 6, 6, 6, and 4. Most reviewers acknowledged the strong performance of the proposed methods and the innovative integration of perturbation-driven forensic clues with the SAM2 architecture. However, several key concerns were raised regarding the empirical validation of its continual learning claims, the robustness of its perturbation-based approach under different conditions, and the usability of images generated by AutoEditForge. In response, the authors provided substantial clarifications and additional experiments, including incremental fine-tuning results to demonstrate adaptive capabilities, throughput comparisons to justify efficiency claims, and saliency visualizations to illustrate adapter mechanisms.

Overall, the proposed Detective SAM framework combined with AutoEditForge shows promising results and addresses important challenges in image forgery localization. Therefore, I recommend acceptance.

**Reviewer Concerns:**

The main concerns of all the reviewers are listed as below:

- Continual learning ability of Detective SAM

    Reviewer sKpy and EVaR raise concerns regarding the initial framing of Detective SAM as a "continual learning" pipeline. sKpy points out that the terminology suggests online, streaming adaptation capabilities, but the experiments only demonstrate offline fine-tuning on small, fixed data batches. A further contradiction is noted between the abstract's implication of handling new editors "without retraining" and the paper's own results showing performance drops on state-of-the-art editors that require fine-tuning. Reviewer EVaR adds that while mitigating catastrophic forgetting is a key motivation, the paper lacks direct evidence that the adapter modules preserve knowledge of older editing distributions after fine-tuning on new ones.

    In response, the authors first admit that "continual" was a misnomer and propose replacing it with a more accurate term (e.g., up-to-date periodic fine-tuning). They clarify that the "training-free" aspect refers specifically to the perturbation-driven forensic signal, not the entire system, whose contribution is *efficient adaptation*via lightweight fine-tuning. To demonstrate stability, they present a new incremental fine-tuning experiment. Regarding catastrophic forgetting, they argue their design mitigates it by keeping the SAM2 backbone frozen and only updating small adapters, and they emphasize that fine-tuning with replay is crucial.

    I think the concerns regarding Reviewer sKpy can be regarded as fully addressed, but only partially addressed for Reviewer EVaR. The results show that replay can help, but forgetting still occurs. This suggests a need for further designs to eliminate catastrophic forgetting, not just mitigate it.

- The design for the perturbations.

    Most reviewers have concerns about the perturbations.

    - Reviewer sKpy concerned that tects "perturbation sensitivity" rather than "forgery semantics" itself. As there are some other types of forgery, such as copy-move or carefully composited human edits, the reviewer questions whether the method is a general image forgery localization framework or a domain-specific solution tailored to forgeries that leave specific perturbation-based artifacts, primarily from diffusion models.  The authors agree that Detective SAM is not focused on semantics and commit to changing their initial wording from "comprehensive framework" to "practical framework" to better reflect its focus on artifacts from modern editors. They justify their approach by arguing that meaningful, harmful modern edits must be semantically consistent, and thus detecting the artifacts left by the generative process is a practical goal. They concede that their focus is on diffusion-based edits and that methods like copy-move are outside their scope, positioning Detective SAM as complementary to semantic-based tools. They also cite literature suggesting perturbation sensitivity might exist in other generative paradigms like GANs. I think the response appear to address this concern by clarifying the main nature of Detective SAM from a practical image forgery detection than a comprehensive ones.
    - Reviewer De2b presents two main points. First, the reviewer questions the method's robustness to confounding factors, specifically whether it can still function if an image is globally post-processed with noise *after* forgery to mask the tampering artifacts. Second, they request a deeper investigation into perturbation combinations, as the justification for selecting "Gaussian Noise + Blur" is insufficient and suggesting experiments with a more diverse set of perturbations. To the first point, the authors conduct a new robustness experiment, applying external perturbations of increasing intensity to their test images. The results indicate that Detective SAM is relatively robust to the types of perturbations it uses as forensic clues. For the second point, the authors expand their ablation study, testing all combinations of their perturbations. The new results show that a three-perturbation combination (JPEG + Noise + Blur) performs best, providing stronger evidence for the impact of perturbation design. Addotionally, they constrained the perturbations to Noise and Blur because of a prior work MINDER, of which argue that each perturbation has a blind spot, e.g. Gaussian blur works best on facial and noise on generic images due to noise being applied uniformly, thus the signal ends up mostly in the background. I think these two concern are resolved by the new provided experiments.
    - Reviewer miFc concerned about the limits of the perturbation strategy, asking if the performance gain from adding more perturbations (N) saturates and encouraging exploration of a wider range of perturbation types. The authors address this by presenting results from testing all combinations of their three perturbations (JPEG, Noise, Blur), showing that the triple combination yields the highest performance. In addition, the authors also explained their choice of Blur and Noise is due to a previous work MINDER. I think this concern is partially addressed, as the saturation beyond the three perturbations is not tested.
- The robustness of the proposed AutoEditForge

    Reviewer De2b and EVaR raise concerns about the robustness and evaluation of the AutoEditForge data generation pipeline. Reviewer De2b focuses on the pipeline's reliability, asking for specific quality control mechanisms, failure rate statistics, and a breakdown of common failure modes. Reviewer EVaR suggests adding an assessment of realism and consistency for the generated edits by AutoEditForge to support the claim that the generated data effectively represents real‑world forgery scenarios. In response to the reviewers, the authors conducted new experiments to validate AutoEditForge's robustness. To address De2b's concern, they performed a failure analysis, and the results demonstrated that their automated quality control mechanisms, specifically duplicate detection and mask validation, can effectively intercept the most common failure modes, thereby ensuring the usability of the generated data. As for the quality assessment requirement raised by Reviewer EVaR, the authors introduced a perceptual quality assessment. By comparing source and tampered images using non-reference metrics, their experiments confirmed that AutoEditForge outputs maintain perceptual fidelity comparable to the original images. With these new added results, the concern about the robustness of the proposed AutoEditForge appears to be addressed.

- The efficiency

    Reviewers sKpy and EVaR raise concerns about the framing of Detective SAM's "lightweight" and "efficient" nature. Reviewer sKpy questions the practical computational efficiency, arguing that the emphasis on a small number of trainable parameters and short training time is misleading. Separately, Reviewer EVaR challenges the fundamental motivation, asking why efficiency is a primary design goal in image forgery localization, a domain that typically prioritizes accuracy.
    In response, the authors directly address the computational cost concern with new empirical data. They conduct a throughput analysis on 512 CoCoGLIDE samples, presenting results that compare inference times across multiple baselines. The data shows that Detective SAM is the third-fastest model, substantiating the "lightweight" claim with what they describe as a "negligible" preprocessing time cost. To address the motivation question, the authors clarify that efficiency is crucial for practical deployment scenarios involving large-scale screening on platforms or resource-limited environments. They justify the design goal by stating that the rapid evolution of generative models necessitates frequent system updates, a process that must be efficient to be practical.

    I believe the authors’ response effectively resolves the concern of Reviewer sKpy by providing quantitative evidence. The response to Reviewer EVaR is adequate but slightly less comprehensive. However, as they successfully argue that efficiency is a necessary component for a usable, up-to-date IFL system, their prioritization of efficiency is reasonable.

**Reviewer Scores:**

After reviewing the paper and the rebuttal, I believe that reviewers sKpy, De2b, and miFc will maintain their positive ratings, with miFc explicitly affirming the paper's quality. For reviewer EVaR, the authors' responses seem to address the main concerns. Therefore, I think EVaR will also result in an improved score.

---

### Decision · Program_Chairs · 2026-01-26

Accept (Poster)